# Factors responsible for *Ixodes ricinus* presence and abundance across a natural-urban gradient

**Thérèse Janzén** *, **Monica Hammer**, **Mona Petersson**, **Patrik Dinnétz**

Södertörn University, School of Natural Sciences Technology and Environmental Studies, Huddinge, Sweden

* therese.janzen@sh.se

**Data Availability Statement:** The data underlying the results presented in the study are available from Swedish national dataservice: https://snd.gu.se/en/describe-and-share-data.

## Abstract

To better understand the spatial distribution of the common tick *Ixodes ricinus*, we investigated how local site factors and landscape characteristics influence tick presence and abundance in different greenspaces along the natural-urban gradient in Stockholm County, Sweden. Ticks and field data were collected in 2017 and 2019 and analyzed in relation to habitat type distributions estimated from land cover maps using geographical information system (GIS). A total of 1378 (992 larvae, 370 nymphs, 13 females, and 3 males) questing ticks were collected from 295 sampling plots in 47 different greenspaces. Ticks were present in 41 of the 47 greenspaces and our results show that both local site features such as vegetation height, and landscape characteristics like the amount of mixed coniferous forest, significantly affect tick abundance. Tick abundance was highest in rural areas with large natural and seminatural habitats, but ticks were also present in parks and gardens in highly urbanized areas. Greenspaces along the natural-urban gradient should be included in surveillance for ticks and tick-borne diseases, including highly urbanized sites that may be perceived by the public as areas with low risk for tick encounters.

## Introduction

Human-environment interactions are crucial components of the epidemiological patterns of vector-borne diseases [1]. The periphery around cities constitutes an environmental gradient ranging from natural and semi-natural areas to highly modified urban landscapes [2]. Natural-urban gradients include a broad variety of greenspaces, spanning from forests to high maintenance parks and gardens [3]. Greenspaces are crucial for terrestrial biodiversity and for human well-being, providing urban citizens with access to nature [4]. Greenspaces also provide suitable habitats for insects and arachnids, like mosquitos and ticks [5]. Unfortunately, these organisms are not only bloodsucking parasites, but they can also jeopardize public health and well-being by transmitting vector-borne pathogens to humans and pets [6–10].

*Ixodes ricinus* (Linnaeus 1758) is a tick species of the family Ixodidae (Acari) commonly found in Europe [11]. *I. ricinus* has three active life stages, larva, nymph, and adult, that each require a blood-meal from a vertebrate host [12]. Humans and our companion animals are potential hosts for all *I. ricinus* life stages. Most tick-borne microorganisms that are pathogenic to humans are predominantly horizontally transmitted and can be found only in nymphs, and

**Funding:** This research was supported by the Foundation for Baltic and East European Studies (https://ostersjostiftelsen.se/en/), grant number 52/18 to PD, MH, MP. The funders had no role in study design, data collection and analysis, decision to publish, or preparation of the manuscript.

**Competing interests:** The authors have declared that no competing interests exist.

adults infected after feeding [13]. Nymphs are more abundant than adult ticks in nature, and due to their smaller size much harder to detect on the human body [14]. Therefore, nymphs are considered the most important stage for the transmission of tick-borne pathogens to humans [15]. The off-host periods for *I. ricinus* are spent in the leaf litter, humus layer, or in the upper soil layers, where they can hide to avoid desiccation and develop to their next developmental stage [16]. During favorable conditions, *I. ricinus* climb up on the lower vegetation and use a "sit-and-wait" strategy called questing until a suitable host passes by [17].

To reduce human risk for tick induced diseases in the future, we need a better understanding of the underlying factors driving tick population dynamics [18]. Local distribution patterns of questing ticks are largely determined by microclimate and availability of hosts, two factors strongly associated with habitat type [19]. Earlier studies have shown that local habitat conditions like tree density and shrub encroachment significantly increased tick abundance, and vegetation height significantly decreased tick abundance [20–22]. At landscape scale, earlier research has shown that the distribution of different forest types affects tick abundance [23–25]. Different land cover types can also affect tick host species abundance, which has a corresponding effect on tick abundance [26]. Many of the earlier studies on tick distributions focused either on small scale local factors, or on large scale landscape characteristics. There have only been a few attempts to bring the different scales together [27, 28].

The aim of this study is to investigate potential factors responsible for *I. ricinus* presence and abundance in different greenspaces along the natural-urban gradient in Stockholm County, Sweden. Specifically, we test whether variation in *I. ricinus* presence and abundance can be explained by local habitat factors, land cover characteristics and landscape configuration related to different levels of urbanization.

## Materials and methods

### Study area

Stockholm County is situated in the hemi-boreal forest zone, has a large archipelago in the Baltic Sea, and a high coverage of natural and semi-natural areas (Fig 1). The predominant vegetation type is coniferous forest, but mixed stands with conifers and broadleaved trees are regularly present. Pure broadleaved stands are mainly found close to arable land and pastures. Most forests are moist, and wetlands are scattered throughout the County [29]. The Stockholm region is one of the fastest expanding urban areas in Europe, applying a polycentric development pattern with regional urban cores. Stockholm is planned with ten green wedges stretching from rural areas to more central parts of the city and many residential areas are therefore interconnected with greenspaces that host a wide variety of plant and animal species [30].

### Sampling site selection and urbanization gradient

In 2017, ticks and field data were collected from 12 different sites around Stockholm County, originally chosen as random controls for a tick-borne encephalitis (TBE) study but never used. In 2019, we made inventories at 35 random sites along the natural-urban gradient around Stockholm. To identify sampling sites along the natural-urban gradient in 2019, we first selected 100 random coordinates within Stockholm County using ArcGIS Pro (Version 2.5.0, ESRI Redland, USA). Using the proportion of *Artificial surfaces* surrounding each site we calculated an urbanization index for each of the 100 coordinates. The urbanization index was calculated for a buffer zone with 1000m radius around each coordinate using satellite data maps with 25 m resolution (reference year 2014) for Stockholm County. These satellite images are freely accessible at the Swedish Environmental Protection Agency's website [31]. The urbanization index ranges from 0 to 100 with the value 0 indicating a completely natural or semi-

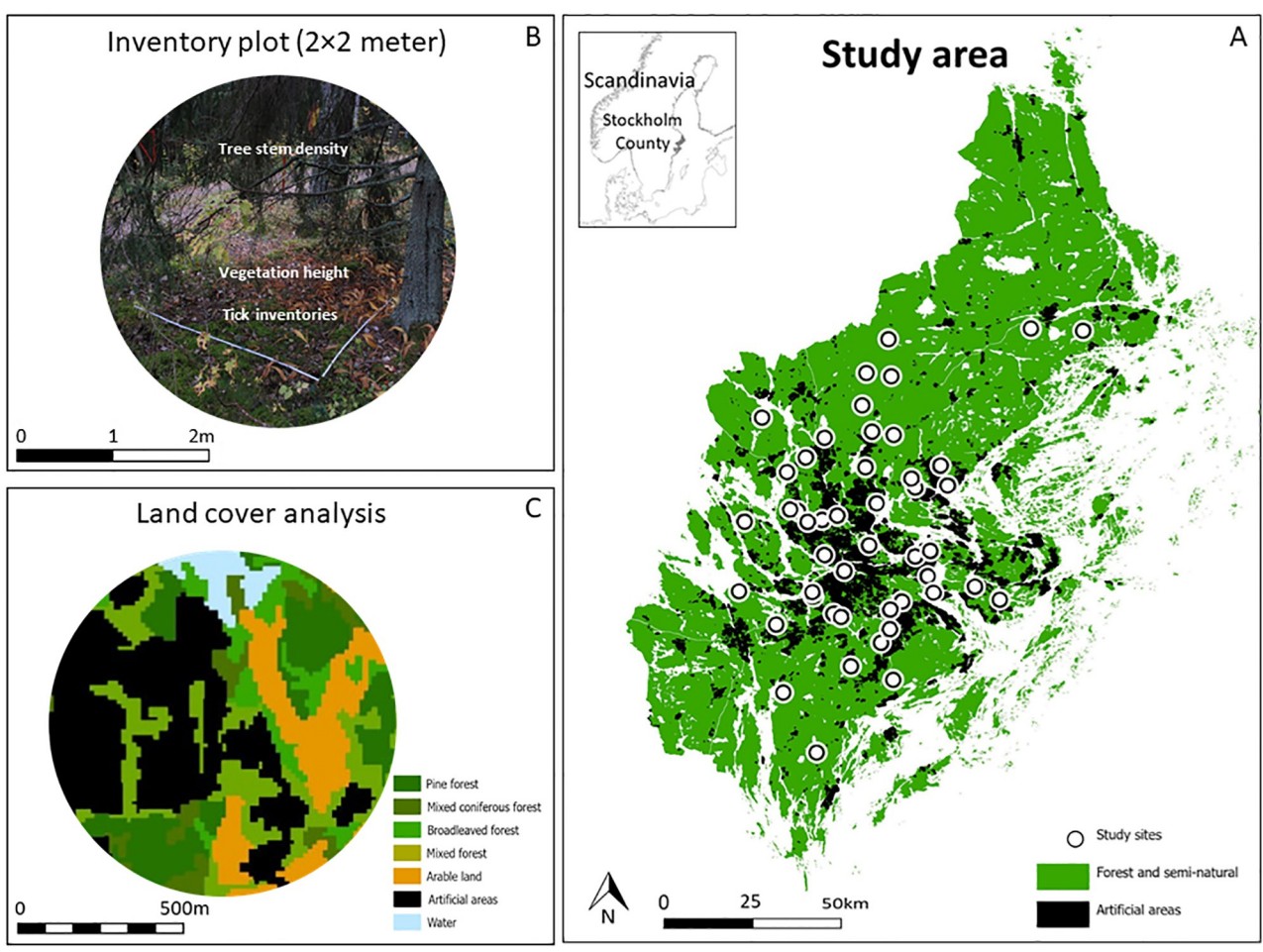

**Fig 1.** The hierarchical study design with: (A) sampling sites randomly distributed along the natural-urban gradient in Stockholm County, Sweden, (B) 2m × 2m sampling plots shown as a photo of one plot at one site, and (C) buffer zones in GIS with a 100m–1000m radius around each sampling site used for land cover data information [31], picture shows a 500m radius buffer. In each sampling plot (B) we collected questing *Ixodes ricinus* and measured the vegetation height, the tree stem density surrounding each plot, and measured the current environmental conditions.

natural site and a value of 100 indicating a landscape with only artificial surfaces. We considered sites with more than 50% *Artificial surfaces* as urban core areas and therefore excluded them from the site selection. Since Stockholm County is surrounded by water, we subtracted the proportion of *Open water* to adjust for presence of non-tick habitats (Eq 1). From all coordinates with an urbanization index ≤ 50% we randomly selected 35 sampling sites. The selected sampling sites ranged from large greenspaces in rural settings, to parks and small patches of vegetation in highly urbanized areas just outside the city center.

$$Urbanization\ index = Artificial\ surfaces\ (\%)\ /\ (100 - Open\ water\ (\%)) \times 100 \qquad (1)$$

At the first 12 sampling sites in 2017, we randomly selected 10 random 2m × 2m plots for inventories of ticks and field data. At the 35 sampling sites in 2019, we collected ticks and field data in 5 random 2m × 2m plots. All 47 sampling sites were visited once, either in 2017 or in 2019, with a total of 295 sampling plots inventoried for ticks and field data. In addition to the random sampling method, we used equal sampling effort in each sampling plot which allows

for comparisons of factors affecting *I. ricinus* presence and abundance among sampling sites as long as we control for the number of plots per site.

## Selection of sampling plots

In south and central Sweden, the current climate allows *I. ricinus* to be active during 6–8 months per year [32]. Nymphs and larvae usually exhibit a bimodal activity pattern with peaks in May—June and August—September, and adult ticks usually exhibit a unimodal activity pattern [33]. Field sampling for both years occurred irregularly with most sampling times in June and August.

The site coordinates randomly selected in GIS served as starting points to select sampling plots within the sampling sites. If any of the coordinates were located on an artificial surface, the nearest greenspace was identified and a random distance from the entrance into the greenspace was chosen to determine a new starting coordinate. For selection of individual 2m × 2m sampling plots at each site we employed a random procedure using compass bearing and distance: starting at the randomly selected GIS generated coordinate, the location for the first sampling plot was determined by holding a compass upside down and randomly rotating the graduation ring to first select a distance between 1m–36m (10˚ on the graduation ring equals 1m), and secondly a random compass direction between 0˚–360˚. The first sampling plot was used as starting point for selecting the second plot, and so on up to five or ten plots per sampling site in 2019 and 2017, respectively. The exact coordinates of each sampling plot were determined with a GPS hand receiver (Garmin eTrex 30).

## Collection of ticks and field data

Tick collections were performed with the mopping technique [34]. The mopping technique entails a white flannel blanket measuring 0.7m × 0.7m attached to a mop. In contrast to the common dragging technique, the mop is held in front of the user and is moved anteriorly over the vegetation. The mop handle permits easy adjustments to different vegetation heights [35]. The total area of each plot was swept once. All questing ticks attaching to the blanket were collected with tweezers, put into tubes, categorized according to life stage, and later stored at -80ºC. Tick life stage categorization was later confirmed in the laboratory.

Seasonal questing activity of *I. ricinus* ticks can differ between biotopes and life stage [14, 33]. To cover unimodal and bimodal activity patterns, ticks and field data were collected from June to October between 10am and 6pm during days without long-lasting precipitation. We sampled on average three sampling sites per day from different levels of urbanization, randomly changing the natural-urban gradient direction on each sampling day. This was done to ensure that sampling at natural, intermediate, and urban areas of the natural-urban gradient was always completed within the same day, and under similar weather conditions. The temperature on the sampling days ranged from 3ºC– 27ºC. For each sampling plot (2017 and 2019) we recorded date, time, temperature, weather conditions, number of ticks, vegetation height, and tree stem density surrounding the plot. Tree stem density was estimated using the Bitterlich sampling technique [36].

## Land cover data

A common procedure for analyzing the effect of different spatial scales is to establish different buffer zone sizes around each sampling location in GIS [37, 38]. To retrieve large scale landscape characteristics, GPS coordinates from the sampling plots were geocoded and incorporated within a GIS. We established 10 buffer zones with increasing radii from 100m to 1000m around the centroid of all sampling plots at each sampling site. The smallest buffer zones

(100m) represent local habitat land cover at the sampling site. Larger buffer zones represent the land cover distribution in the adjacent landscape around each sampling site. To extract the proportion of the different land cover types surrounding the sampling site we used satellite land cover maps from reference year 2019 [39]. These maps have a spatial resolution of 10m and include the following six main categories: 1) *Forest and seminatural areas*, 2) *Open areas*, 3) *Arable land*, 4) *Wetlands*, 5) *Artificial surfaces* and 6) *Inland and marine water*. These main categories are further divided into subcategories with detailed information regarding the different land cover classes (S1 Table). Based on the biological and ecological requirements for *I. ricinus* and to identify landscape risk factors we decided to use the main categories in the analyses, with the exception of *Forest and seminatural areas* where we included eight individual forest types: *Pine forest*, *Spruce forest*, *Mixed coniferous forest*, *Mixed forest*, *Broadleaved forest*, *Broadleaved hardwood forest*, *Broadleaved forest with hardwood forest* and *Temporarily non-forest* (S1 Table).

## Landscape configuration

To calculate landscape configuration metrics at landscape level surrounding each sampling site, we used land cover data from GIS buffers with a radius of 1000m, exported to GeoTIFF format. Buffer zones with a radius of 1000m were used to examine the landscape effects in the adjacent landscape around each sampling site and to include the home ranges of preferred hosts for *I. ricinus* [40]. Landscape configuration metrics for all 47 sampling sites were estimated with FRAGSTATS version 4 [41]. For landscape heterogeneity we used *Shannons' diversity index* (*SHDI*), estimating the diversity of all included land cover classes at each sampling site weighted by their proportional coverage. *Contagion* (*CONTAG*) of land cover types was used to measure the aggregation of landscape attributes that can influence the suitability of sites for different tick host species. To analyze the influence of forest configuration on tick abundance, we collapsed all forest types into one main forest category. As measures of forest configuration, we used *percent of forest cover* (*PLAND*) and *total forest edge* length (*TE*).

## Statistical methods

All statistical analyses were performed with R version 4.0.3 [42]. To analyze the effect of possible risk factors for tick abundance in different greenspaces across the natural-urban gradient, we used generalized linear mixed models assuming Poisson distributed residuals. As the data contained a larger proportion of zeros than would be expected according to a Poisson or a negative binomial distribution causing overdispersion [43, 44], we fitted zero-inflated Poisson models using the package *glmmTMB* (generalized linear mixed models using Template Model Builder) [45].

The *glmmTMB* package allows for handling of the correlative structure among sampling units within sites and has two main parts: (1) a conditional model that reports the coefficients of the Poisson count model, and (2) a zero-inflated model that reports the probability of a fixed effect resulting in the observation of an extra zero that is not generated by the conditional model [45]. The zero-inflation probability is bounded between zero and one by using a logit link. The zi-formula is a binomial model turned upside down, meaning that a positive coefficient indicates a higher chance of absence. The zero-inflated models deal with the common problem that the mechanisms determining presence/absence data can be different from those that determine abundance. The conditional models incorporate the hierarchical structure that exists in different landscapes by allowing for site specific land cover relationships. The site specificity in the models also accounts for the spatial correlation that may exist in the relationship between ticks and their environment as well as for the occurrence of a larger number of

zeros than the standard Poisson distribution can incorporate. Since tick abundance was expected to be affected by factors measured at multiple spatial scales, our models include both sampling plot conditions, local site factors and large landscape characteristics.

In the full models, tick abundance was analyzed as a function of the proportion of each land cover type. Larvae and nymphs were analyzed in two separate models due to the clustered occurrence of larvae throughout the landscape, and to the large difference in importance as disease vector. Due to very low n-value adult ticks were excluded from the statistical analyses. In both models, the fixed factors *Pine forest*, *Spruce forest*, *Mixed coniferous forest*, *Mixed forest*, *Broadleaved forest*, *Broadleaved hardwood forest*, *Broadleaved forest with broadleaved hardwood forest*, *Temporarily not forest*, and the main categories *Open areas*, *Arable land*, *Wetlands*, *Artificial surfaces*, and *Inland and Marine waters*, together with the local site conditions, *Vegetation height*, *Tree stem density* and *Temperature* were included. The second order polynomial of *Month* indicating sampling time (sampling *Day* caused convergence issues) was added to correct for the irregular sampling times. In addition, sampling site was used as a random variable to adjust for the difference in sampling effort between sites, and to handle the correlative structure among plots within the same sampling site in the conditional part of the model. Model selection was performed by stepwise backward model reduction of fixed factors [43] using p-values and Akaike´s Information Criterion (AIC) [46]. The second order polynomial of sampling *Month* was retained in all models even if the linear and quadratic terms were nonsignificant in the final model. The same procedure was applied separately to the conditional component and to the zero-inflated component of the model. The same modeling procedure was repeated for each buffer zone size in 100m increments ranging from 100m–1000m radius.

Tick abundance was also analyzed as a function of landscape configuration in generalized linear mixed zero-inflated models with the degree of *Urbanization*, and the landscape configuration variables *PLAND*, *TE*, *CONTAG*, and *SHDI* as fixed landscape scale factors and *Vegetation height*, *Tree stem density* and *Temperature* as fixed local scale factors. We also included the second order polynomial of *Month* as a temporal fixed effect, and study *Site* as random factor. We again made separate models for larvae and for nymphs and excluded adult ticks from the analyses. Model selection was performed similarly as for the models analyzing the effects of land cover types. However, in the landscape configuration models, data was only analyzed for buffer zones with 1000m radius. Since *TE* will be zero when forest cover is 0 or 100% and peak at intermediate forest levels, it was difficult to disentangle the forest edge effect in different greenspaces. To account for this, we also included a two-way interaction between *PLAND* and TE.

All models were tested for multicollinearity using the variance inflation factor (VIF) [47, 48]. To provide an indication of the final model's goodness-of-fit, we estimated the variance explained by the fixed factors (marginal $R^2$) and the variation explained by both the fixed and random factors (conditional $R^2$). Model fit was assessed by the residual distribution of each model using package DHARMa [49].

## Results

Ticks of the species *Ixodes ricinus* were present in 41 of the 47 inventoried greenspaces. In total, we collected 1378 ticks, including 992 larvae, 370 nymphs, 13 adult females and 3 adult males. The tick life stage capture ratio for larvae:nymphs:adults was 100:31:1.6. Adult ticks were very few and therefore excluded from the statistical analyses. High VIF values indicating multicollinearity resulted in removal of the landscape factors *Pine forest*, *Temporarily non-forest*, and *Open areas* in the 800m radius buffer for nymph abundance, and the landscape configuration factor *CONTAG* was removed from the analyses of landscape configuration effects for

nymph abundance. Conditional $R^2$ was higher than marginal $R^2$ (Tables 1 and 2), indicating that a substantial part of the explained variance of presence and abundance is due to differences among sampling sites not accounted for by the fixed factors. The conditional and the zero-inflated model parts provided slightly different patterns for the fixed factors that were responsible for *I. ricinus* presence and abundance, respectively. In general, sampling plot factors were most important for tick abundance, especially the negative effect from *Vegetation height* (Tables 1–3). Landscape characteristics affected both tick presence and abundance, with negative effects of *Artificial surfaces* and mixed effects from both open land and forests

**Table 1. The effect of local scale factors and landscape variables on *Ixodes ricinus* larvae measured in buffer zones consisting of 100m increments within a 100m–1000m radius.** Presence and abundance of *I. ricinus* larvae were analyzed in generalized linear mixed zero-inflated models. Estimated coefficients and significance levels for the explanatory variables from both the conditional model part (abundance) and the zero-inflated model part (presence/absence) is reported for each buffer zone size. For easier interpretation we have changed the sign of the estimates for the zero-inflated model part meaning that a positive coefficient should be interpreted as a higher likelihood for tick presence.

| Fixed factors | Local site | Surrounding landscape | | | | | | | | |
|---|---|---|---|---|---|---|---|---|---|---|
| | 100m | 200m | 300m | 400m | 500m | 600m | 700m | 800m | 900m | 1000m |
| **Conditional part (abundance)** | | | | | | | | | | |
| *Spruce forest* | | | -0.12* | | | | | | | |
| *Mixed coniferous forest* | | | 0.10* | | | | | | | |
| *Mixed forest* | | | | | | | -0.14* | | | |
| *Broadleaved hardwood forest* | | | | | | | 0.26* | | 0.49** | 0.48** |
| *Open land* | | | 0.12** | 0.12** | | 0.16*** | | 0.19*** | | |
| *Artificial surfaces* | | | -0.11** | -0.10** | | -0.12** | -0.13** | -0.14** | -0.19** | -0.20** |
| *Vegetation height*[a] | -0.03*** | -0.03*** | -0.03*** | -0.03*** | -0.03*** | -0.03*** | -0.03*** | -0.03*** | -0.03*** | -0.03*** |
| **Zero-inflated part (presence/absence)** | | | | | | | | | | |
| *Spruce forest* | -0.05* | | | | | | | | | |
| *Mixed forest* | | | | | | | 0.26** | | | |
| *Broadleaved forest* | | | 0.13** | | | | | -0.49** | | |
| *Broadleaved hardwood forest* | | | 0.22*** | 0.24*** | 0.37*** | 0.29*** | 0.20** | 1.07*** | 0.27** | |
| *Broadleaved forest with broadleaved hardwood forest* | | | | | 0.01** | | | | | |
| *Temporarily non-forest* | | | 0.37** | 0.38*** | 0.31** | | | 0.30* | | |
| *Wetland* | | | | | | | | 0.35* | | |
| *Open land* | -0.09*** | | -0.13*** | 0.14*** | -0.17*** | -0.17*** | -0.15** | -0.28*** | -0.15** | |
| *Artificial surfaces* | | | | | -0.15* | | | | | |
| *Water* | | | | | 0.08* | | | 0.13** | 0.05* | - |
| *Tree stem density*[a] | | | 0.08* | 0.06* | 0.07* | | | | | |
| *Temperature* | | | | | -0.28* | | | | | |
| *Month* | | | | | | | | 0.98** | | |
| *Month²* | | | | | | | | -0.68** | | |
| Random factor | | | | | | | | | | |
| Site n | 47 | 47 | 47 | 47 | 47 | 47 | 47 | 47 | 47 | 47 |
| Marginal $R^2$ | 0.236 | 0.195 | 0.041 | 0.319 | 0.275 | 0.024 | 0.267 | 0.289 | 0.395 | 0.320 |
| Conditional $R^2$ | 0.566 | 0.548 | 0.534 | 0.442 | 0.411 | 0.501 | 0.629 | 0.455 | 0.635 | 0.545 |
| Plot n | 295 | 295 | 295 | 295 | 295 | 295 | 295 | 295 | 295 | 295 |

Significance levels:

* $P<0.05$;

** $P<0.01$;

*** $P<0.00$

[a] Local plot factor

**Table 2. The effect of local scale and landscape scale variables on *Ixodes ricinus* nymphs measured in buffer zones consisting of 100m increments within a 100m–1000m radius.** Presence and abundance of *I. ricinus* nymphs were analyzed in generalized linear mixed zero-inflated models. Estimated coefficients and significance levels for the explanatory variables from both the conditional model part (abundance) and the zero-inflated model part (presence/absence) is reported for each buffer zone size. For easier interpretation we have changed the sign of the estimates for the zero-inflated models meaning that a positive coefficient should be interpreted as a higher likelihood for tick presence.

| | Local site | Surrounding landscape | | | | | | | | |
|---|---|---|---|---|---|---|---|---|---|---|
| | 100m | 200m | 300m | 400m | 500m | 600m | 700m | 800m | 900m | 1000m |
| **Fixed factors** | | | | | | | | | | |
| **Conditional part (abundance)** | | | | | | | | | | |
| *Spruce forest* | | | -0.07* | -0.10** | -0.07* | | | | | |
| *Mixed coniferous forest* | | | 0.08** | 0.08** | 0.10*** | 0.07* | 0.10* | | | 0.10* |
| *Broadleaved hardwood forest* | | | -0.07* | | | | | | | |
| *Broadleaved forest with broadleaved hardwood forest* | -0.17* | | | | -0.26* | | | | | |
| *Artificial surfaces* | | -0.05** | | -0.08*** | | | | | | |
| *Vegetation height*[a] | -0.01** | -0.01** | -0.01*** | -0.01*** | -0.01** | -0.01*** | -0.01** | -0.01** | -0.01*** | -0.01** |
| **Zero-inflation part (presence/absence)** | | | | | | | | | | |
| *Pine forest* | | | | | -0.04* | -0.07* | | | -0.07* | -0.08* |
| *Spruce forest* | | | | | | -0.23* | | -0.12** | -0.18* | -0.18* |
| *Mixed coniferous* | | | | | | | -0.14* | | | |
| *Broadleaved forest* | 0.12** | | | | | | | | | |
| *Broadleaved hardwood forest* | | | | | | | 0.16* | | 0.51* | |
| *Broadleaved forest with broadleaved hardwood forest* | | -0.20* | | | | | | | -0.66* | - |
| *Temporarily non-forest* | | | | | | | | | | -0.26* |
| *Open land* | -0.06* | | | | | | | | | |
| *Water* | | | | | 0.18** | 0.11* | 0.10** | | | |
| *Artificial surfaces* | | -0.06** | -0.10*** | | -0.11*** | -0.20*** | -0.12*** | -0.13** | -0.16** | -0.26*** |
| *Temperature* | | | | | -0.21* | | | | | |
| *Month* | | | | | 10.28** | | | | | 5.69* |
| *Month²* | | | | | -0.73** | | | | | -0.41* |
| **Random factor** | | | | | | | | | | |
| Site n | 47 | 47 | 47 | 47 | 47 | 47 | 47 | 47 | 47 | 47 |
| Marginal R² | 0.114 | 0.194 | 0.228 | 0.356 | 0.261 | 0.120 | 0.087 | 0.035 | 0.043 | 0.131 |
| Conditional R² | 0.573 | 0.577 | 0.517 | 0.580 | 0.554 | 0.538 | 0.512 | 0.537 | 0.526 | 0.192 |
| Plot n | 295 | 295 | 295 | 295 | 295 | 295 | 295 | 295 | 295 | 295 |

Significance levels:

* P<0.05;

** P<0.01;

*** P<0.00

[a] Local plot factor

depending on tick life stage and type of forest (Tables 1–3). In addition, there was a significant curvilinear effect of the factor *Month* with a peak in late summer for abundance of both larvae and nymphs (Tables 1 and 2).

## Sampling plot factors

The local site factor *Vegetation heigh* had significant negative effects on both larva and nymph abundance, indicating that we have fewer questing *I. ricinus* in plots where the field layer vegetation is high (Fig 2). Ninety percent of all the ticks in this study were collected in vegetation heights of 30cm or lower. *Tree stem density* surrounding sampling plots had a significant

**Table 3. The effect of urbanization and landscape configuration on questing *Ixodes ricinus* presence and abundance measured in 1000m radius buffer zones analyzed in generalized linear mixed zero-inflated models.** Estimated coefficients and significance levels for the explanatory variables from both the conditional (abundance) and zero-inflated (presence/absence) model components are reported for *I. ricinus* larvae and nymphs. Parameter estimates and significance levels for landscape factors are estimated based on a 1000m radius buffer zone and for local site factors from individual sampling plots. For easier interpretation we have changed the sign of the estimates for the zero-inflated models meaning that a positive coefficient should be interpreted as a higher likelihood for tick presence and vice versa.

| Fixed factors | Larvae | Nymphs |
|---|---|---|
| **Conditional part (abundance)** | | |
| *Vegetation height*[a] | - | -0.01** |
| **Zero-inflated part (presence/absence)** | | |
| *Urbanization* | - | -0.10*** |
| *Tree stem density*[a] | 0.06* | - |
| *PLAND* | 0.05* | - |
| **Random factor** | | |
| Sites n | 47 | 47 |
| Marginal $R^2$ | 0.167 | 0.040 |
| Conditional $R^2$ | 0.564 | 0.535 |
| Plots n | 295 | 295 |

Significance levels:

*P<0.05;

** P<0.01;

*** P<0.00

[a] Local plot factor

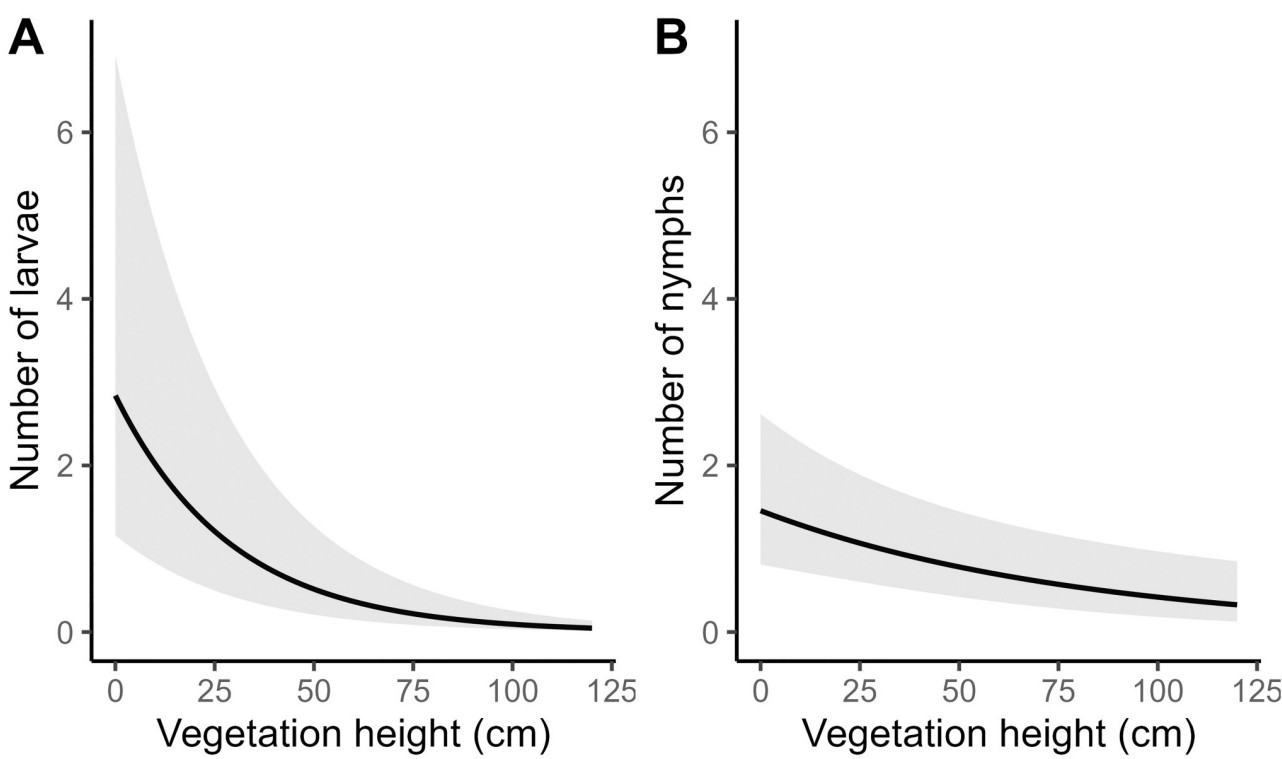

**Fig 2. Effect of vegetation height on number of *Ixodes ricinus* larvae (A) and nymphs (B).** The results from the glmmTMB show significant negative effects of field layer vegetation height with more questing *I. ricinus* in lower vegetation. Grey area indicates 95% confidence band.

positive effect on the presence of *I. ricinus* larvae, but no significant effect on the presence of nymphs (Tables 1–3). *Temperature* had a weak significant negative effect on the presence of both larvae and nymphs (Tables 1 and 2). In our study ticks were actively questing in temperatures ranging from 3°C– 27° but most larvae and nymphs (95%) were collected at temperatures between 15°C– 24°C.

### Local site factors

At local scale, increasing proportions of *Open land* at the sampling sites had a significant negative effect on both larva and nymph presence (Tables 1 and 2). There was also a significant negative effect of the habitat factor *Spruce forest* on larva presence (Table 1). A high proportion of *Broadleaved hardwood forest* had a significant positive effect on nymph presence, and *Broadleaved forest* stands had a significant negative effect on nymph abundance (Table 2).

### Surrounding landscape characteristics

Our results show significant landscape effects on both presence and abundance of *I. ricinus*. Large proportions of monocultures of *Spruce forests* and *Pine forests* in the largest buffer zones had a significant negative effect on nymph presence, and a significant negative effect on nymph abundance in intermediate buffer zones (Tables 1 and 2). On the other hand, forest stands with *Mixed coniferous forest* in intermediate and large buffer zones showed significant positive effects on nymph abundance (Tables 1 and 2). The effects from all types of coniferous forests in the landscape on larva distribution did not show any obvious patterns.

Presence and abundance of *I. ricinus* larvae were instead significantly positively affected by *Broadleaved forest* and *Broadleaved hardwood forest* (Tables 1 and 2). The effects of broadleaved forest types on nymphs were varied or lacking. It is important to point out that the overall proportion of broadleaved forest in Stockholm County is low, still the significant effect on larvae is evident. *Temporarily non-forested* habitats showed a significant positive effect on larva presence.

The proportion of *Open land* in the landscape had no effect on nymphs (Table 2) but a significant negative effect on the presence of larvae (Table 1). However, in sampling plots where larvae were present the proportion of *Open land* in the landscape had a significant positive effect on larva abundance (Table 1). The proportion of *Water* in intermediate sized buffer zones had a significant positive effect both on larva and nymph presence (Tables 1 and 2). Finally, *Artificial surfaces* had a significant negative effect on larva abundance (Table 1), and nymph presence (Table 2) in almost all buffer zone sizes.

### Urbanization and landscape configuration

The effects of landscape configuration and urbanization on tick abundance were estimated from the 1000m radius buffer zones surrounding the different sampling sites. *Urbanization* had a significant negative effect on the presence of nymphs (Table 3). However, even if fewer ticks were found in more urbanized areas, ticks were present also in areas with 30–40% urbanization (Fig 3). The percent forest in the landscape *(PLAND)* had significant positive effects on the presence of larvae (Table 3). Neither the landscape configuration metrics *SHDI*, *CONTAG*, *TE* nor the two-way interaction between *PLAND* and *TE* had significant effects on tick abundance or tick presence.

A simplified summary of the main result from all analyses focusing on the consistent patterns displayed by the effects of fixed local and landscape factors is shown in Table 4.

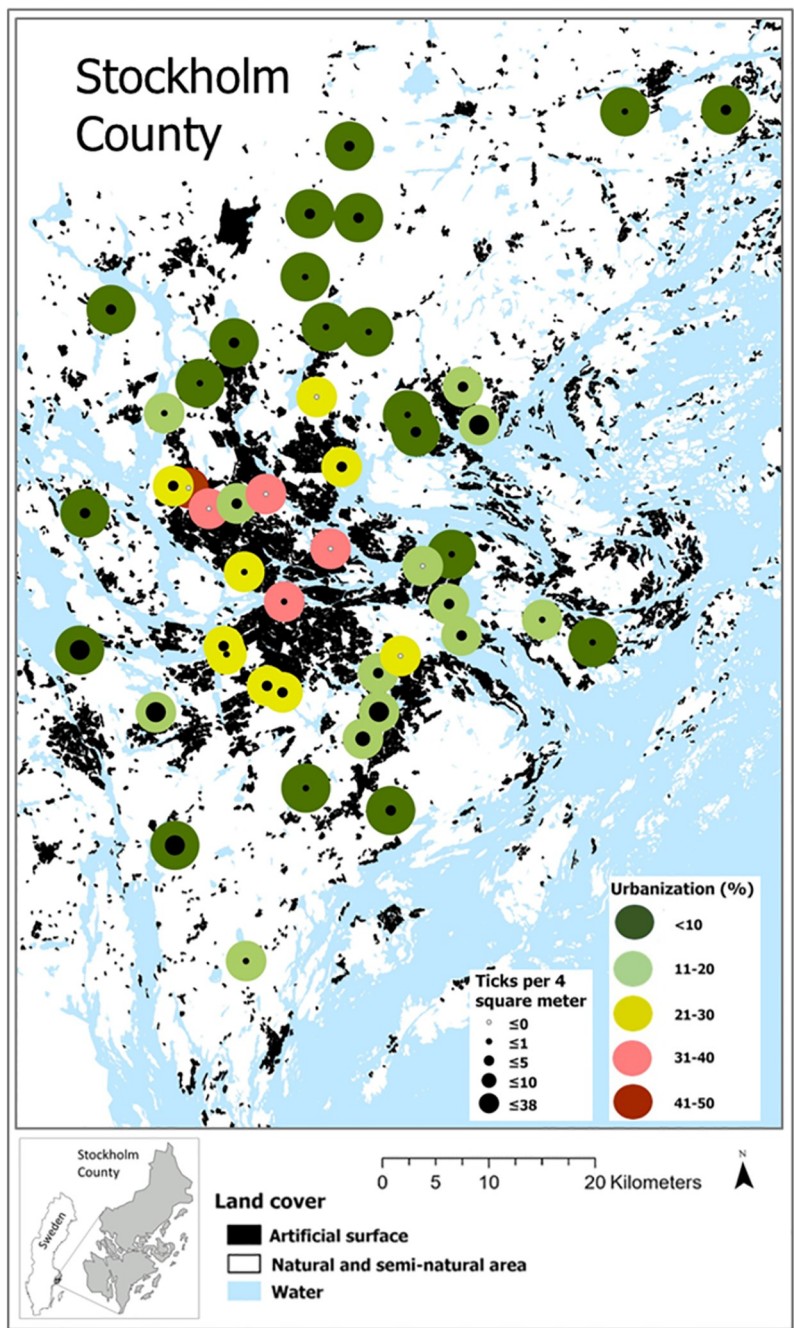

**Fig 3. The degree of urbanization and *Ixodes ricinus* abundance at each sampling site in Stockholm County.**

## Discussion

To analyze factors responsible for *Ixodes ricinus* distribution, we investigated the effects from sampling plot characteristics together with local sampling site factors and effects from the surrounding landscape at 47 different sites along the natural-urban gradient in Stockholm County, Sweden. One of the main findings is that sampling plot features mainly are important for variation in tick abundance among sampling sites, while landscape features are important

**Table 4. Summary of main results from Tables 1–3.** Factors responsible for *Ixodes ricinus* presence and abundance in greenspaces across the urbanization gradient. Sampling plot conditions include recordings at each site and local site factors include land cover types in the smallest GIS buffer zone (100m radius). Landscape features include land cover types (200-1000m radius) and landscape configuration variables (1000m radius).

| Fixed factors | Larvae | Nymphs |
|---|---|---|
| **Sampling plot conditions** | | |
| *Vegetation height* | Negative | Negative |
| *Tree stem density* | Positive | - |
| **Local site factors** | | |
| *Coniferous forest (monoculture)* | Negative | - |
| *Open land* | Negative | Negative |
| **Landscape features** | | |
| *Artificial surfaces* | Negative | Negative |
| *Coniferous forest (monoculture)* | Negative | Negative |
| *Mixed coniferous forest* | - | Positive |
| *Broadleaved hardwood forest* | Positive | Positive |
| *Urbanization* | Negative | Negative |
| *PLAND* | Positive | - |

for the occupancy pattern of *I. ricinus* across sampling sites, but also for tick abundance at sites were *I. ricinus* is present.

## Local plot level

Local microhabitat characteristics are important for tick survival and reproduction [50–53]. Ticks are sensitive to desiccation and need to hide from the sun during warm weather conditions [54]. Most of the ticks collected during our study were found in vegetation lower than 30cm, confirming earlier findings showing that vegetation height has a significant negative effect on tick presence. While this could be a technical field sampling effect, the mopping technique we used in our study allowed us to move the blanket both above, beneath, and in between plants, even in high and dense field layers, reducing the risk of missing ticks in hard-to-reach field layers. Interestingly, the sampling plots with the highest tick abundance were found in forests with almost no field vegetation, and only a thin layer of leaf litter.

*I. ricinus* mainly move vertically up and down straws and stems in the field layer, risking desiccation if trapped high up in the vegetation [55]. Previous studies have shown that ticks are questing close to the top of the vegetation, preferably at heights between 25cm– 50cm. Vegetation taller than 75cm is used less for questing, probably due to the desiccation risk [56]. An alternative hypothesis is that high vegetation is a perfect habitat for finding hosts and that plots with high vegetation are already emptied of ticks by large mammal hosts before we conduct our sampling. The latter is a less likely but not impossible scenario and will require experimental studies to be resolved. However, the high abundance of *I. ricinus* in low vegetation increases the risk for tick infestation of humans because we often avoid tall grass and shrubs and instead prefer areas and paths with low vegetation height that are easier to traverse when we are out hiking.

Previous studies consistently found that *I. ricinus* appear to reduce their questing activity in response to extreme daily temperatures [57–59]. In our study, most ticks (95%) were collected in temperatures between 15º C to 24º C, indicating that this temperature range is beneficial for questing, especially for larvae. However, ticks were actively questing from 3º C to 27º C,

throughout the study period. The statistical analysis did not reveal any consistent temperature effect, though we rarely encountered extreme temperatures during sampling. The main problem during tick sampling is rain or extremely damp conditions, therefore no sampling was conducted during days with long periods of rain.

In this study, *I. ricinus* larvae were often collected in clusters. After being hatched from aggregations of eggs, larvae tend to be found in a clumped distribution. Considering life cycle stages, tick larvae are assumed to be most sensitive to local site conditions, and have the highest mortality rate [60]. This pattern may apply for *I. ricinus* in many environments, but it was not supported by the tick life stage capture ratio for larvae:nymphs:adults in the present study. However, our study was not optimized to measure the life stage capture ratio, as we used a standardized sampling strategy designed to compare differences among different sites and habitats. Nymphs and adults are known to quest close to the location where individual ticks dropped off from their previous host [61]. Nymphs and adults were mostly found in a scattered pattern which likely represents their host-based distribution pattern.

## Local site factors

Land cover types in the smallest buffer zone (100m) represent the local habitat surrounding the inventoried plots. Sampling sites with a large proportion of open land in the smallest buffer zone had significantly lower presence of both larvae and nymphs. We interpret this as a direct negative effect of open land on both tick presence due to drier microclimatic conditions and tick host presence. Previous studies found that trees shadowing the sampling plots may lower the risk of desiccation for *I. ricinus* [21], corresponding with our findings of significant positive effects of tree stem density around the sampling plots on the presence of larvae. The positive effect of tree stem density in the local habitat surrounding each plot may have been counteracted by the significant negative effects of dense monocultures of spruce forest or pine forest. More favorable local site conditions for ticks seemed to be present at sampling sites located in mixed coniferous forests or in broad leaved hardwood forests. Coniferous forest is a very common forest type around Stockholm especially at the outer edges of the nature-urban gradient. Broadleaved hardwood forests are rare around Stockholm and more common in highly maintained park environments in peri-urban and urban environments.

## Landscape level

In this study, greenspaces with surrounding degree of urbanization ranging from 0%– 50% were included in the sampling. The results show significant negative effects of artificial surfaces and urbanization on tick abundance, with higher tick abundance in greenspaces surrounded by large natural and semi-natural habitats. However, we also found ticks in parks and small pockets of vegetation in highly urbanized areas. Previous research and our own observations confirm that urban greenspaces can provide favorable ecological conditions for both small and large vertebrate hosts [5, 6, 62]. This suggests that ticks repeatedly are being transported to these habitats by wildlife, and that ticks can have access to hosts capable of feeding all tick stages also in urban environments [8, 63–65]. In Stockholm County, large green wedges traverse the entire natural-urban sector like spokes in a wheel, leading from the rural rim to the city hub. The proportion of forest within these wedges is substantial in many areas and there are established populations of prime tick hosts such as rodents, birds, rabbits, hares, foxes, and roe deer even in the most densely human populated areas [66].

Variation in tick abundance between greenspaces differing in habitat characteristic and location has been well documented [25, 67, 68]. Forest patches affect the survival of ticks by creating a humid microhabitat as well as determining habitat suitability for hosts. Broadleaved

forests and mixed forests harboring a diverse fauna are generally considered ideal habitats for *I. ricinus* [69]. However, coniferous forests in areas with high rainfall and with a thick moist litter layer can also support high tick densities [70, 71]. The most important land cover type in the surrounding landscape promoting *I. ricinus* abundance in different green-spaces around Stockholm is mixed coniferous forest. *I. ricinus* was also present in landscapes with high proportion of broadleaved hardwood forests which is in agreement with previous studies [61, 67]. However, in Stockholm County broadleaved hardwood forest is extremely rare except for highly maintained parks and small remnants of the old farming landscape. The structures of the surrounding landscape influence *I. ricinus* abundance through variation in abiotic conditions and tick-host dynamics [72]. Despite the importance of these different land cover variables on tick abundance, site variation also played an important role, indicating that there are additional factors influencing *I. ricinus* abundance along the urbanization gradient.

In addition to landscape composition, landscape configuration may also affect tick abundance along the natural-urban gradient. Percent forest cover had a positive effect on larval presence, which is probably connected to the environmental conditions affecting density and activity of *I. ricinus* main hosts [56]. One of the forest configuration factors that we analyzed was the total length of the forest edge. Forest edges have been suggested as important tick habitats due to their suitability for many tick hosts [73]. In this study, we could not see any effect of forest edge, neither as an interaction factor with forest area nor as a main factor on its own. Similarly, there was no clear effect of forest aggregation on tick presence or abundance, which in contrast to earlier studies [25, 74]. The lack of forest configuration effect may be explained by the green wedges stretching all the way from rural to urban areas. Due to these wedges, the proportion of forest in buffer zones with a 1km radius is substantial in many areas of Stockholm County, even close to the city center. However, even if the forest configuration and composition does not change dramatically along the natural-urban gradient the proportion of artificial surfaces is very different.

The natural-urban gradient represents many special ecological characters affecting both ticks and hosts [75]. The presence of animal hosts for all active stages and the ability to disperse within and between habitats are important for the life cycle of *I. ricinus* [76]. Nymphs, the medically most important tick stage, were found along the entire natural-urban gradient, but adult ticks were absent in highly urbanized greenspaces. Urbanized greenspaces surrounded by built environments typically have reduced diversity of wildlife, especially larger mammals on which adult ticks usually feed [6]. The few adult ticks sampled in this study were found in suburban forests and at these same sites, the other tick life stages were present as well. Suburban forests seem to have favorable conditions for all tick stages and options for finding suitable hosts for their blood meals. Future studies should include an assessment of the occurrence and density of wildlife at the different collection sites which might shed more light on the large variation in tick presence and abundance between our study sites.

## Factors responsible for *Ixodes ricinus* presence and abundance

Our study demonstrates the presence of *I. ricinus* in greenspaces across the entire natural-urban gradient in Stockholm County, but with higher abundance of ticks in intermediate urbanized areas and large natural and seminatural greenspaces. Tree abundance surrounding each plot had a significant positive effect on the presence of larvae, and forest patches consisting of mixed coniferous forest or broadleaved hardwood forest in the surrounding landscape significantly increased both larval and nymphal presence and abundance. In many parts of Europe, climate change is altering the growth and development of forest types, with a decline

of coniferous forests and an increase in mixed and broad-leaved forests. A change in forest composition and structure might increase the proportion of habits suitable for ticks in this region, especially larvae, which might consequently influence the epidemiology of tick-borne diseases [21].

Higher tick abundance will increase the risk for human tick encounters. However, the range of vegetation types where we found ticks included greenspaces with very low vegetation height, and highly urbanized parks. This information is relevant to public health, since humans and their pets are more likely to spend more time, and make closer skin contact, with low vegetation in parks or along trails than with tall rough vegetation in large and remote greenspaces [71]. In a polycentric city like Stockholm with green wedges stretching along the urbanization gradient, ticks are present in a wide variety of greenspaces, even at sites often perceived to be free of tick hazards by the public.

## Conclusions

This study investigated *Ixodes ricinus* presence and abundance in relation to patterns of urbanization, and simultaneously considered small scale local environmental conditions, and large-scale landscape characteristics. Our systematic random sampling method can be used for comparisons among sites within a study as well as with other studies using similar procedures. The main finding from our study is that even if there is a higher abundance of *I. ricinus* in natural and semi-natural areas compared to urban areas, ticks are still present at highly urbanized sites. We also found that *I. ricinus*, along the entire natural-urban gradient, is readily abundant in sites with very low field vegetation height. At landscape scale, we found that in the hemi-boreal forest zone around Stockholm in Sweden, mixed coniferous forest seems to be an important driver for *I. ricinus*. The presence of *I. ricinus* in urban areas and its preference for low vegetation height makes the probability for human tick encounters high also in urban areas. Based on our results, we recommend greenspaces in urban areas to be included in surveillance for ticks and tick-borne diseases.

## Supporting information

**S1 Table. Swedish land cover nomenclature including grid code, land cover type and definition.**
(DOCX)

## Acknowledgments

The authors thank the editor and the anonymous reviewers for their useful input. In addition, thanks to Kari Lethilä for help with statistical models and Raphaela Mayerhofer for help with editing the manuscript.

## Author Contributions

**Conceptualization:** Thérese Janzén, Monica Hammer, Mona Petersson, Patrik Dinnétz.

**Data curation:** Thérese Janzén, Mona Petersson, Patrik Dinnétz.

**Formal analysis:** Thérese Janzén, Mona Petersson, Patrik Dinnétz.

**Funding acquisition:** Monica Hammer, Mona Petersson, Patrik Dinnétz.

**Investigation:** Thérese Janzén, Mona Petersson, Patrik Dinnétz.

**Methodology:** Thérese Janzén, Patrik Dinnétz.

**Project administration:** Thérèse Janzén, Patrik Dinnétz.

**Supervision:** Monica Hammer, Mona Petersson, Patrik Dinnétz.

**Visualization:** Thérèse Janzén, Patrik Dinnétz.

**Writing – original draft:** Thérèse Janzén.

**Writing – review & editing:** Thérèse Janzén, Monica Hammer, Mona Petersson, Patrik Dinnétz.

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
