## [Decision Letter · Decision Letter 0]

11 Sep 2022

PONE-D-22-15726Risk factors for tick exposure across an urbanization gradientPLOS ONE

Dear Dr. Janzen,

Thank you for submitting your manuscript to PLOS ONE. After careful consideration, we feel that it has merit but does not fully meet PLOS ONE’s publication criteria as it currently stands. Therefore, we invite you to submit a revised version of the manuscript that addresses the points raised during the review process.

 As you'll read, reviewers were split with respect to the value of your manuscript.  Given these split perspectives, my editorial decision is to allow you opportunity to potentially address the primary concern of Reviewer 1, which deals with the skewed representation of life stages in your collection and how that limits applicability of your data to any of your findings and general assessment of 'risk'.   Additional concerns:Both reviewers were not convinced that 'risk' had actually been assessed. Both also thought that sampling was insufficiently described.The Results section could be more meaningfully organized to increase general understanding/relevance.Note, given the split perspectives of the reviewers and "major revision" designation, the manuscript will go back out for review after a revised version is submitted.   Please submit your revised manuscript by Oct 26 2022 11:59PM. If you will need more time than this to complete your revisions, please reply to this message or contact the journal office at plosone@plos.org. Please include the following items when submitting your revised manuscript:A rebuttal letter that responds to each point raised by the academic editor and reviewer(s). You should upload this letter as a separate file labeled 'Response to Reviewers'.A marked-up copy of your manuscript that highlights changes made to the original version. You should upload this as a separate file labeled 'Revised Manuscript with Track Changes'.An unmarked version of your revised paper without tracked changes. You should upload this as a separate file labeled 'Manuscript'.

We look forward to receiving your revised manuscript.

Kind regards,

Dr. Janice L. Bossart

Academic Editor

PLOS ONE

Journal Requirements:

2. Please amend your manuscript to include your abstract after the title page.

4. We note that Figure 1 and 3 in your submission contain [map/satellite] images which may be copyrighted. All PLOS content is published under the Creative Commons Attribution License (CC BY 4.0), which means that the manuscript, images, and Supporting Information files will be freely available online, and any third party is permitted to access, download, copy, distribute, and use these materials in any way, even commercially, with proper attribution. For these reasons, we cannot publish previously copyrighted maps or satellite images created using proprietary data, such as Google software (Google Maps, Street View, and Earth). For more information, see our copyright guidelines: http://journals.plos.org/plosone/s/licenses-and-copyright.

    1. You may seek permission from the original copyright holder of Figures 1 and 3 to publish the content specifically under the CC BY 4.0 license.  

Reviewers' comments:

Reviewer's Responses to Questions

**Comments to the Author**

1. Is the manuscript technically sound, and do the data support the conclusions?

Reviewer #1: Partly

Reviewer #2: Yes

2. Has the statistical analysis been performed appropriately and rigorously? 

Reviewer #1: Yes

Reviewer #2: Yes

3. Have the authors made all data underlying the findings in their manuscript fully available?

Reviewer #1: Yes

Reviewer #2: Yes

4. Is the manuscript presented in an intelligible fashion and written in standard English?

Reviewer #1: Yes

Reviewer #2: Yes

5. Review Comments to the Author

Reviewer #1: Line 101: Note that "mopping" is modified dragging technique, as dragging is much better known.

Line 109: I did not see any details about the frequency of collections from each site per season. How frequently were sites visited per year? The large number of larvae and relatively small number of adults (lines 198 - 199) could indicate undersampling during peak nymph/adult activity.

Lines 250 - 285: A lot of information is provided here that recites what can be read in the tables. Written text here would be best used for highlighting the patterns to help draw the reader's eye to particular areas of the table. There are several ways I can suggest to consider organizing the text: 1) organize by broad general patterns related to land cover (e.g. lines 244 - 245, mixed forest) vs. unusual patterns (e.g. lines 257 - 258, broadleaved with hardwood, and 268 - 269, wetlands) or 2) organize by positive effects vs. negative effects or 3) organize by strength of the association, i.e. coefficients. The text explanation of these results should help aid in understanding or highlighting parts of the tables presented.

Line 324 -326: Vegetation height for adults in particular could be biased by a small collection (n = 13). Much more variation might be seen if more adults were collected.

Line 377: Is this a reference to the wetlands results? Please clarify

Spelling/Grammar/Word usage

Line 322: "tick's" should be changed to "ticks"

Line 330: grass

Line 332: change "is indicating" to indicates

Line 342: represents

Line 372: Revise this sentence, "absolutely most ticks" is awkward

Line 388: measures

Line 406 - 407: Incomplete sentence

Line 412: compared

Final note: Given the author's stated interest in tick-borne disease surveillance, life stages relevant to transmission should be included in larger numbers. Typically, researchers look at density of infected nymphs (DIN) to discuss human risk of exposure to disease. As the authors noted, the clustering of larvae interfered with the analysis, but larvae are also over represented in this study especially given that this is a life stage that generally poses less disease risk to humans and animals.

Limited collection of adults (n = 13) compared to larvae (n = 992) is a concern because some major tick-borne disease, e.g. Borrelia spp. infections, are typically transmitted by adults and nymphs only. Maternal transmission of Borreliae to eggs/larvae does not occur, so unfed larvae are not considered a disease risk for Borrelia. Thus, targeting adult and nymph life stages in collections is imperative to understand the whole system.

Although the authors here are not considering infection in ticks, I recommend looking more closely at literature in which DIN is calculated and used to interpret exposure risk.

In one paper, Jouda et al. 2004, the authors collected and analyzed 108 - 145 nymphs per altitude class, per year, with a total of 1112 nymphs included in the study. Likewise, 9 - 149 adults were included per altitude class, per year, for a total of 507 adults across three years. The authors should seek to collect comparable numbers, particularly when evaluating such a large collection of environmental variables for their effect on densities.

I applaud the authors for the spatial study design and analyses. The limited and skewed tick collections, unfortunately undermine the results overall, and the applicability of this data to understand human exposure risk.

Reviewer #2: The overall value of presented study is high and the outcome may have additional value in tick-borne diseases ecoepidemiology and use as starting point in landscape planning and urban planning related studies, risk measurements for public health. The strength of the manuscript and presented study is solid basis of up-to date literature, with substantial representation of papers describing local environment characteristics that allowed for good study design. Manuscript is not free from minor redaction issues and few major gaps, yet this should be easily improved by the Authors.

Title: please specify which ticks! Also, In Reviewer's opinion it's abour risk factors, not risks in it's current state.

Line no. – comment

9-10 – broad thesis, please provide examples of epidemiological studies

9-17 – Phrases like: „. Greenspaces of all kinds are crucial for terrestrial biodiversity” or „unwanted insects” or „are not only annoying” sounds maybe catchy for lay-man, but are biased, empty and not suitable for scientific journal.

Thus I strongly advise to review and adjust the language used.

15 „arachnoids” (arachnids).

17 – what inf. Dis.? This statement lacks of examples of vector borne diseases of epidemiological importance for precision.

19 – how long, what is long in tick lifecycle?

24 – rephrase (ticks surely are not hoping for anything)

34-38 – in reviewer’s opinion it is important to underline that these descriptions are for I. ricinus, because other ticks of medical importance, Dermacentor reticulats, have different (almost opposite) patterns of occurence, described i.a. in vast environmental studies performed by Mierzejewska et al.

40 – like above, specify which ticks. It is worth to specify in discussion as well.

43-46 – please ellaborate importance of nymphs, within introduction, before the aims of the project. As for now such insert is confusing.

56 – latin radius, please try rephrase using a singular nom. for clarity throughout the ms.

Fig 1. Please state if inventory plot 2x2 is an example photography or satellite view. It is unclear (not easily distinguishable what is presented in the circle picture). Maybe arrows with description would be helpful.

80 – please provide reference to urbanization index

98 – is this the description of the snowball technique? Could you please confirm if that means that the compass arrow pointed both distance between 0-36 m as 1degree to 10 cm, as well as the direction? This randomization technique seems interesting and I feel it was not covered enough.

101-106 – please state if were collected ticks cathegorized at site or later?

107-112. In line 88 you stated that there were 35 Sites monitored by checking five 2x2 inventory plots. In lines 107-112 you stated that sampling were performed in 8h period of the day for ALL 35 sites at the same day, is that correct? Or do you mean that in each Site sampling was performed within same day? How many timeas each Site was sampled? Please clarify sampling as this is not clear for now, consider adding some suppl. materiały e.g tables.

299 – this is probably Fig 3 description?

General comment:

Aim of the study was to investigate the risk of tick encounters (not done?)

The tool was to identify the potential risk factor (success).

Table 4? I think this is actually risk factors, not risks. How were those risks valued? Is it solely on the strength of effects? Or is there any formula used?

303: risk factors rather than risk, suggestion: it is worth considering explaining in methods that you intended to evaluate risk on the basis of tick presence/abundance in reference to any detected risk factors. For now ms is lacking risk measurement description, it is only resulting from discussion part and it's virtually about risk factors.

406-407 - for rephrasing, seem like different kind of journalism

It seems logical that risk is affected by positive or negative effects, however risk itself was not described in my opinion. E.g., what is the matching value for statement that "there is still a considerable risk for tick encounters in highly urbanized areas.". What risk level is considerable and what is not?

Hopefully these comments will be of use for imprevements of the manuscript.

6. PLOS authors have the option to publish the peer review history of their article (what does this mean?). If published, this will include your full peer review and any attached files.

Reviewer #1: No

Reviewer #2: No

---

## [Author Response · Author response to Decision Letter 0]

21 Oct 2022

PLOS ONE

October 21, 20222

Re: Factors responsible for Ixodes ricinus abundance across an urbanization gradient

PONE-D-22-15726

Dear reviewers,

We have read your comments carefully and tried our best to address them one by one. Please see answers your comments below. All modification in the manuscript have been highlighted in yellow.

Kindest regards,

Corresponding author

Thérese Janzén

PhD student

therese.janzen@sh.se

Phone +46 8 608

Södertörn University, School of Natural Sciences, Technology and Environmental Studies

SE-141 89 Huddinge, Sweden

Responses to reviewer #1 comments:

Line 101: Note that "mopping" is modified dragging technique, as dragging is much better known.

Response: We have added information reading our mopping technique to the manuscript (lines 146-149). 

Line 109: I did not see any details about the frequency of collections from each site per season. How frequently were sites visited per year? The large number of larvae and relatively small number of adults (lines 198 - 199) could indicate undersampling during peak nymph/adult activity.

Response: We have added details regarding the frequency of our collections to the manuscript and explanations to the capture ratios for the different life stages (lines 154-158).

Lines 250 - 285: A lot of information is provided here that recites what can be read in the tables. Written text here would be best used for highlighting the patterns to help draw the reader's eye to particular areas of the table. There are several ways I can suggest to consider organizing the text: 1) organize by broad general patterns related to land cover (e.g. lines 244 - 245, mixed forest) vs. unusual patterns (e.g. lines 257 - 258, broadleaved with hardwood, and 268 - 269, wetlands) or 2) organize by positive effects vs. negative effects or 3) organize by strength of the association, i.e. coefficients. The text explanation of these results should help aid in understanding or highlighting parts of the tables presented.

Response: Thank you very much for pointing this out. We have revised this section of the result section. We took you advice and organized by broad general patterns with focus on the effects of different land cover types on occurrence and abundance of Ixodes Ricinus (lines 291-333). 

Line 324 -326: Vegetation height for adults in particular could be biased by a small collection (n = 13). Much more variation might be seen if more adults were collected.

Response: Our systematic random sampling model with data collected from many different sites produced data on presence and absence of ticks which is crucial information when analyzing factors responsible for I. ricinus distribution. This sampling method yields mean number of ticks per four square meter which allows for comparisons between sites, within this study as well as with other studies using similar procedures. In addition, in the statistical analyses adults and nymphs were analyzed together (n=317+13=330).

Line 377: Is this a reference to the wetlands results? Please clarify

Response: Thank you for your question. We have edited this sentence and clarified that we are discussing the land cover type Open water (lines 431-433).

Line 322: "tick's" should be changed to "ticks"

Response: Revised accordingly (line 372).

Line 330: grass

Response: Revised accordingly (line 380). 

Line 332: change "is indicating" to indicates

Response: Thank you for pointing this out. We have revised this sentence (line 384-385).

Line 342: represents

Response: Revised accordingly (line 392). 

Line 372: Revise this sentence, "absolutely most ticks" is awkward

Response: Thank you very much for pointing this out. We have revised this sentence (line 406-407). 

Line 388: measures

Response: We have revised this sentence (lines 441-443). 

Line 406 - 407: Incomplete sentence

Response: Thank you very much for pointing this out. We have revised this sentence (lines 460-462).

Line 412: compared

Response: Revised accordingly (line 470).

Final note: Given the author's stated interest in tick-borne disease surveillance, life stages relevant to transmission should be included in larger numbers. Typically, researchers look at density of infected nymphs (DIN) to discuss human risk of exposure to disease. As the authors noted, the clustering of larvae interfered with the analysis, but larvae are also over represented in this study especially given that this is a life stage that generally poses less disease risk to humans and animals.

Limited collection of adults (n = 13) compared to larvae (n = 992) is a concern because some major tick-borne disease, e.g. Borrelia spp. infections, are typically transmitted by adults and nymphs only. Maternal transmission of Borreliae to eggs/larvae does not occur, so unfed larvae are not considered a disease risk for Borrelia. Thus, targeting adult and nymph life stages in collections is imperative to understand the whole system.

Although the authors here are not considering infection in ticks, I recommend looking more closely at literature in which DIN is calculated and used to interpret exposure risk.

In one paper, Jouda et al. 2004, the authors collected and analyzed 108 - 145 nymphs per altitude class, per year, with a total of 1112 nymphs included in the study. Likewise, 9 - 149 adults were included per altitude class, per year, for a total of 507 adults across three years. The authors should seek to collect comparable numbers, particularly when evaluating such a large collection of environmental variables for their effect on densities.

I applaud the authors for the spatial study design and analyses. The limited and skewed tick collections, unfortunately undermine the results overall, and the applicability of this data to understand human exposure risk.

Response: We were pleased about the kind words about our spatial study design and analyzes. We believe that this sampling method is important both for comparison between sites as well as with other studies using similar procedures. 

In our spatial analysis design, tick absence is as important as tick presence both for assessing the prevalence of ticks along the urbanization gradient and to understand how different habitats effects the occurrence of ticks in different areas. 

Responses to reviewer #2 comments: 

The overall value of presented study is high and the outcome may have additional value in tick-borne diseases ecoepidemiology and use as starting point in landscape planning and urban planning related studies, risk measurements for public health. The strength of the manuscript and presented study is solid basis of up-to date literature, with substantial representation of papers describing local environment characteristics that allowed for good study design. Manuscript is not free from minor redaction issues and few major gaps, yet this should be easily improved by the Authors.

Response: We also believe that the overall value of our study is high and with the comments and suggestions from both reviewers we hope that the manuscript will be accepted for publication. 

Title: please specify which ticks! Also, In Reviewer's opinion it's about risk factors, not risks in it's current state.

Response: Thank you very much for pointing this out. We have changed the aim of the paper from risks to possible factors responsible for Ixodes ricinus abundance across an urbanization gradient in Northern Europe (lines 67-68). 

9-10 – broad thesis, please provide examples of epidemiological studies

Response: We have added the following reference to this statement (line 30):

Vanwambeke SO, Linard C, Gilbert M. Emerging challenges of infectious diseases as a feature of land systems. Current Opinion in Environmental Sustainability. 2019 Jun 1;38:31-6.

In this article, the authors are discussing how various environmental processes including land cover and land use affects infectious diseases and especially vector-borne diseases. Land cover and land use connects vectors, hosts and humans, including different activities exposing them to pathogens. 

Human-environment interactions are therefore crucial components of the epidemiological patterns of vector-borne diseases. 

9-17 – Phrases like: „. Greenspaces of all kinds are crucial for terrestrial biodiversity” or „unwanted insects” or „are not only annoying” sounds maybe catchy for lay-man, but are biased, empty and not suitable for scientific journal.

Thus I strongly advise to review and adjust the language used.

Response: Thank you very much for pointing this out. We have revised this section (lines 34-37). 

15 „arachnoids” (arachnids).

Response: Revised accordingly (line 35).

17 – what inf. Dis.? This statement lacks of examples of vector borne diseases of epidemiological importance for precision.

Response: We don’t think so since vector-borne diseases are not the focus of this paper. To be clearer we have revised this sentence and added references (lines 35-37).

19 – how long, what is long in tick lifecycle?

Response: Thank you for your question. We revised this sentence (lines 39-40).

24 – rephrase (ticks surely are not hoping for anything)

Response: Thank you for this nice reminder. We have revised this sentence (lines 49-50).

34-38 – in reviewer’s opinion it is important to underline that these descriptions are for I. ricinus, because other ticks of medical importance, Dermacentor reticulats, have different (almost opposite) patterns of occurence, described i.a. in vast environmental studies performed by Mierzejewska et al.

Response: Thank you for pointing this out. We have revised the manuscript and underlined that these descriptions are for I. ricinus ticks (lines 56-60).

40 – like above, specify which ticks. It is worth to specify in discussion as well.

Response: Revised accordingly.

43-46 – please ellaborate importance of nymphs, within introduction, before the aims of the project. As for now such insert is confusing.

Response: Thank you for pointing this out. We have added information regarding the distribution of I. ricinus life stages and importance of nymphs to the introduction (lines 41-47). 

56 – latin radius, please try rephrase using a singular nom. for clarity throughout the ms.

Response: Revised accordingly

Fig 1. Please state if inventory plot 2x2 is an example photography or satellite view. It is unclear (not easily distinguishable what is presented in the circle picture). Maybe arrows with description would be helpful.

Response: Thank you very much for pointing this out. We have stated that is a photography from one of our collection sites and added descriptions to what is presented in the picture. 

80 – please provide reference to urbanization index

Response: A common way of calculating an urbanization index is to use the proportion of artificial areas in a surrounding area. We have in addition, extracted open water since Stockholm County is surrounded by water (lines 111-120). 

98 – is this the description of the snowball technique? Could you please confirm if that means that the compass arrow pointed both distance between 0-36 m as 1degree to 10 cm, as well as the direction? This randomization technique seems interesting and I feel it was not covered enough.

Response: Thank you very much for your nice comment. We have added more information about our systematic random sampling technique to the method section (lines 137-145).

101-106 – please state if were collected ticks cathegorized at site or later?

Response: Thank you for your nice reminder. We have added to the manuscript that the ticks were categorized in the laboratory as well (line 151-152). 

107-112. In line 88 you stated that there were 35 Sites monitored by checking five 2x2 inventory plots. In lines 107-112 you stated that sampling were performed in 8h period of the day for ALL 35 sites at the same day, is that correct? Or do you mean that in each Site sampling was performed within same day? How many timeas each Site was sampled? Please clarify sampling as this is not clear for now, consider adding some suppl. materiały e.g tables.

Response: Thank you very much for your questions. We sampled about three sites per day from different gradients of urbanization, either it the direction from more rural to urban or from more urban to rural. We have added information regarding our sampling method to this section of the manuscript (lines 127-131, 154-158).

299 – this is probably Fig 3 description?

Response: Revised accordingly (line 348).

Aim of the study was to investigate the risk of tick encounters (not done?)

The tool was to identify the potential risk factor (success).

Response: Thank you very much for pointing this out. We have changed our aim to possible factors responsible for I. ricinus presence and abundance across an urbanization gradient.

Table 4? I think this is actually risk factors, not risks. How were those risks valued? Is it solely on the strength of effects? Or is there any formula used?

Response: Thank you very much for pointing this out. We have changed from ‘risk factors’ to ‘factors responsible for I. ricinus abundance across an urbanization gradient’.

303: risk factors rather than risk, suggestion: it is worth considering explaining in methods that you intended to evaluate risk on the basis of tick presence/abundance in reference to any detected risk factors. For now ms is lacking risk measurement description, it is only resulting from discussion part and it's virtually about risk factors.

Response: We have explained in the methods section that we analyzed possible risk factors for I. ricinus presence and abundance (lines 67-68, 77-78, Table 4). 

406-407 - for rephrasing, seem like different kind of journalism

Response: Thank you for this kind reminder. We have revised this sentence (lines 460-462). 

It seems logical that risk is affected by positive or negative effects, however risk itself was not described in my opinion. E.g., what is the matching value for statement that "there is still a considerable risk for tick encounters in highly urbanized areas.". What risk level is considerable and what is not?

Response: Thank you very much for pointing this out. We have changed from ‘risk factors’ to ‘factors responsible for I. ricinus abundance across an urbanization gradient’.

We hope that our manuscript is now ready for publication in PLOS ONE.

---

## [Decision Letter · Decision Letter 1]

7 Dec 2022

PONE-D-22-15726R1Factors responsible for *Ixodes ricinus* presence and abundance across an urbanization gradient

PLOS ONE

Dear Dr. Janzen,

Thank you for submitting your manuscript to PLOS ONE. After careful consideration, we feel that it has merit but does not fully meet PLOS ONE’s publication criteria as it currently stands. Therefore, we invite you to submit a revised version of the manuscript that addresses the points raised during the review process.

Between the original and revised version, your manuscript has now been assessed by three external reviewers.  All feel the main focus, an investigation of ticks in urban areas and the risks they pose to humans and pets, has value.  That said, reviewers have also identified major limitations of the study, e.g. experimental design, sampling effort, and the fact so few adults were collected.  These limitations have not been adequately addressed or embraced in the revised version with respect to data interpretation and conclusions that can be drawn.  It is also unclear how exactly sampling was conducted across sites (and time) and whether the approach was adequate for assessing tick activity over the period of the study.  

Given the mixed reviews (from minor revision to reject outright), I'm willing to let the manuscript progress through another round of revision and review.  Please take reviewer concerns fully to heart, otherwise similar concerns will likely continue to be raised by the reviewer community.  Alternatively, you may decide you would prefer submitting your manuscript to another, more focused, journal.

We look forward to receiving your revised manuscript.

Kind regards,

Janice L. Bossart

Academic Editor

PLOS ONE

Reviewers' comments:

Reviewer's Responses to Questions

**Comments to the Author**

1. If the authors have adequately addressed your comments raised in a previous round of review and you feel that this manuscript is now acceptable for publication, you may indicate that here to bypass the “Comments to the Author” section, enter your conflict of interest statement in the “Confidential to Editor” section, and submit your "Accept" recommendation.

Reviewer #3: (No Response)

2. Is the manuscript technically sound, and do the data support the conclusions?

Reviewer #3: Partly

3. Has the statistical analysis been performed appropriately and rigorously? 

Reviewer #3: I Don't Know

4. Have the authors made all data underlying the findings in their manuscript fully available?

Reviewer #3: No

5. Is the manuscript presented in an intelligible fashion and written in standard English?

Reviewer #3: Yes

6. Review Comments to the Author

Reviewer #3: This is a useful investigation into what local site and landscape level factors influence the abundance of Ixodes scapularis in an urban landscape. Importantly, this study highlights the fact that ticks can occur in urbanized areas, and thus may pose a risk to human and companion animal health. Nevertheless, there are several major and minor shortcomings of this study.

The major issues include:

The use of the term “rural-urban gradient”. This term does not accurately reflect the landscape variables tested in this study. Non-urban (aka. artificial surface) doesn’t necessarily mean rural.

This study is better suited to just look at nymphs and larvae. The limited collection of adult ticks makes it difficult to make any conclusions about the adult life stage.

The outcome of objective 2 (effect of land cover at each sampling site) is not clear. The methods state all sampling sites were in greenspaces, but do not provide landcover information.

It seems that there was no repeat sampling of sites over time. This is a major (though not fatal) flaw in study design. To convince readers of the relevance of this data set, greater effort is needed to explain how time was factored into the modeling framework. Furthermore, the distance sampled per site (40m2 if I did my calculations correctly) falls well below sampling efforts in most studies (~1000m2 per site).

There is no explanation in the methods and results on how the response variable (abundance) was adjusted to account for the variation in sampling effort across sites. For example, density of ticks should be calculated per site and then this metric would be comparable across sites with different plot numbers.

Reanalysis and interpretation of the results is warranted prior to publication.

Unable to find data available on the stated website. Please provide direct link.

The minor issues are:

Line 21: describe what “40% urbanization” is in the context of this study.

Line 72: effect of local factors on what? effect of land cover on what? Results and discussion talk about presence/absence of ticks, but this not stated in any objectives.

Line 79: remove “large” – this is a vague qualifier

Line 84: the area of a circle with a 100m radii would be 0.0314km2

Line 107: At what time of year did sampling occur in 2017? How many times were sites visited/resampled? What were the intervals of collection? How does this relate to known activity time of life stages in the region?

Line 127: At what time of year did sampling occur in 2019? How many times were sites visited/resampled? What were the intervals of collection? How does this relate to known activity time of life stages in the region?

Line 128: Were any of the sites sampled in 2017 resampled in 2019? It is unclear.

Line 129: Okay. So, this statement makes it sound like each site was only sampled once over the entire duration of the study. That means a site consists of sampling from a single time point and 5 or 10 4m2 plots. This totals, at most, 40m2 of sampling per data point. While there is no authority over what the minimum sampling effort for ticks should be, this seems very low and may explain why so few adults were collected over the course of the study.

Line 135: It is unclear what local landcover the sampling occurred in. It is just referred to as greenspace here, but was it forest and if so what type? This is central to the second objective of the study.

Line 138: What is the snowball technique? Is that what is described in lines 138-144?

Line 154: How do you “cover” the activity patterns of ticks by only sampling each site a single time and having those single sampling occasions span from May to October?

Line 160: what is meant by the “height of the field layer”? what distance around the plot was used for stem density measurements? Was vegetation height/tree cover also measure in both 2017 and 2019, or only 2019?

Line 179: include list of the eight forest subcategories here

Line 182: This size of the forest fragments that sampling sites were in would be an important variable to include in this analysis.

Line 184: Justification for why 1000m buffers were used for urbanization and landscape composition models is needed. One could perform an analysis to determine the appropriate ‘scale of effect’.

Line 185: Was SHDI and contagion measured using the landcover classification with or without the forest subcategories?

Line 193: do you mean “To analyze the effect of possible risk factors….”?

Line 210: It is unclear how spatial autocorrelation is accounted for within the models. Please explain and/or show results of tests for spatial autocorrelation.

Line 216: What aspect/measure of each land cover type is being used? Proportion?

Line 216: How was this variation in sampling effort (5 vs 10 plots) accounted for in your response variable of abundance?

Line 219: Including the proportion of all landcover types in a single model is likely not appropriate as explanatory variables are very likely to be highly correlated given that the data is proportional. While the authors state that they tested for collinearity, these results are not reported, and the full models probably violated this rule. Please show VIF results and/or reduce variables included in full model.

Line 239: Need justification for why counts of nymphs and adults were combined. Because few adults were collected, it seems reasonable to just run a model for larvae and a model for nymphs.

Line 277: Could the fewer ticks in taller vegetation be a detection issue?

Table 1: Why so little consistency in significant landscape variables across scales for larval presence/absence and abundance?

Line 449: It is confusing to have a discussion of tick abundance under a section title tick presence and absence.

Figure 3: Add an inset map for spatial context and explain the difference between the white and black in the background.

7. PLOS authors have the option to publish the peer review history of their article (what does this mean?). If published, this will include your full peer review and any attached files.

Reviewer #3: No

---

## [Author Response · Author response to Decision Letter 1]

20 Jan 2023

PLOS ONE

January 21, 2023

Re: Factors responsible for Ixodes ricinus presence and abundance across a natural-urban gradient

PONE-D-22-15726R1

Dear Janice L. Bossart,

thank you for the opportunity to revise our manuscript. We appreciate you and the reviewers for your time in reviewing our paper and providing valuable comments. We have considered the comments and tried our best in addressing every one of them. Please see answers to all comments below. All modification in the manuscript have been highlighted in yellow.

Kindest regards,

Corresponding author

Thérese Janzén

PhD student

therese.janzen@sh.se

Phone +46 8 608 4000

Södertörn University, School of Natural Sciences, Technology and Environmental Studies

SE-141 89 Huddinge, Sweden

Response to main comments 

We agree with the reviewers that our manuscript on ticks in urban areas and the risk they pose to humans are pets are important to investigate. The comments and suggestions from the editor and the reviewers have been a great help in improving our manuscript. We have started with answering the main comments raised from the editor. 

Our sampling methods were designed to estimate the presence and abundance of Ixodes ricinus along the natural-urban gradient in Stockholm County, Sweden. First, regarding our experimental design we have made changes and clarifications throughout the manuscript. Second, regarding the difference in sampling effort (5 or 10 plots) we have here as well, clarified our methods and explained that we are analyzing tick presence and abundance in the individual plots how the factor Site have been used in the statistical models to adjust for the difference in sampling effort at the different sites. 

Third, regarding the number of adults collected in this study it is our understanding that in the natural tick populations, there are much fewer adults compared to larvae and nymphs (Dobson et al., 2011; Ruiz-Fons et al., 2012). This is due to the longevity of I. ricinus ticks and that a large amount of time is spent of off the host, where the environment is posing a threat of survival of individual ticks (Mejlon, 2000). We have removed the adults from all the statistical models and the focus is now on the presence and abundance of larvae and nymphs. In addition, we have clarified in the manuscript how sampling was conducted across sites and time. 

Our revised manuscript is now clearer! 

We have also included a point-by-point response to the reviewer´s specific comments below.

Dear reviewer,

We have read your comments carefully and tried to address them one by one. Please see answers to your comments below. All modification in the manuscript have been highlighted in yellow.

Kindest regards,

Corresponding author

Thérese Janzén

PhD student

therese.janzen@sh.se

Phone +46 8 608 4000

Södertörn University, School of Natural Sciences, Technology and Environmental Studies

SE-141 89 Huddinge, Sweden

Response to Reviewer #3 comments

This is a useful investigation into what local site and landscape level factors influence the abundance of Ixodes scapularis in an urban landscape. Importantly, this study highlights the fact that ticks can occur in urbanized areas, and thus may pose a risk to human and companion animal health. Nevertheless, there are several major and minor shortcomings of this study.

Response: We also believe that this is an important investigation into both local site and landscape level factors influencing tick presence and abundance in urban landscape. We have addressed all of the comments which helped us improving our manuscript. 

The use of the term “rural-urban gradient”. This term does not accurately reflect the landscape variables tested in this study. Non-urban (aka. artificial surface) doesn’t necessarily mean rural.

Response: Thank you for pointing this out. We have revised the terminology and changed from rural-urban to natural-urban gradient.

This study is better suited to just look at nymphs and larvae. The limited collection of adult ticks makes it difficult to make any conclusions about the adult life stage.

Response: The adult ticks have been excluded from the statistical analyses and the focus is now on the effect of land cover composition and configuration on larval and nymphal tick presence and abundance. 

The outcome of objective 2 (effect of land cover at each sampling site) is not clear. The methods state all sampling sites were in greenspaces, but do not provide landcover information.

Response: Since we have sampled ticks in random greenspaces along the urbanization gradient, sampling sites are variable and can therefore consist of e.g., forests, meadows, open areas, or urban parks. For local land cover information at each site, we have analyzed the proportion of the different land cover types present in the smallest buffer zone (100m).

It seems that there was no repeat sampling of sites over time. This is a major (though not fatal) flaw in study design. To convince readers of the relevance of this data set, greater effort is needed to explain how time was factored into the modeling framework. Furthermore, the distance sampled per site (40m2 if I did my calculations correctly) falls well below sampling efforts in most studies (~1000m2 per site). 

Response: In this study we have made a trade off between the number of sampling sites and the number of sampling plots per site. To be able to have a larger n-value for sites we have a used 5 or for some sites 10 replicates of sampling plots within each site. This will increase the statistical power for explaining among site variation but the representativity of each site is of course lower than if we have had more plots per site. Our argument for using 2m × 2m plots is that we want to mop the plot with our blanket and then quickly remove ticks from the blanket before they drop off by themselves. For this the 2m × 2m plot is an ideal size. This plot size is also crucial for the possibility to measure the very local conditions like vegetation height, tree steam density. Even if tree steam density is covering a slightly larger are it always start from the sampling plot which will be impossible if the sampling area is too large. This will again be a trade off between optimal tick sampling, vegetation height and tree steam density measurements. The chosen plot size is optimal for bringing all of this together.

There is also a trade-off between resampling individual sites and the number of sites inventoried. We have chosen to use many sites but only visit them once. Instead, we use a second order polynomial factor of sampling time as a covariate to account for the sampling time effect. We have sampled sites at different positions along the natural-urban gradient throughout the whole sampling time frame. In this way we control the effects from our other fixed factors for time. We also regarded the sampling time to be important and have therefore kept the time factors in all final models. 

There is no explanation in the methods and results on how the response variable (abundance) was adjusted to account for the variation in sampling effort across sites. For example, density of ticks should be calculated per site and then this metric would be comparable across sites with different plot numbers.

Response: In the statistical models, tick presence and abundance are estimated in each sampling plot. By using sampling sites as random factor in the mixed models we adjust for the correlative structure of plots within site and control the repetitive measurements at each site. This will allow us to compare sites even if they have different numbers of sampling plots. All mean values analyzed are calculated at the 2m × 2m plot size. 

Reanalysis and interpretation of the results is warranted prior to publication.

Response: We have reanalyzed all the statistical models with only larvae and nymphs. In addition, we have reinterpreted the results and made changes accordingly in the discussion. 

Unable to find data available on the stated website. Please provide direct link.

Response: The data will be provided when the manuscript has been accepted for publication (as recommended on the PLOS ONE website).

Line 21: describe what “40% urbanization” is in the context of this study.

Response new line 21: We have revised the sentence.

Tick abundance was higher in rural areas with large natural and seminatural habitats, but ticks were also present in parks and gardens in highly urbanized areas.

Line 72: effect of local factors on what? effect of land cover on what? Results and discussion talk about presence/absence of ticks, but this not stated in any objectives.

Response new line 71: Thank you for pointing this out. We have reformulated the objectives of this study.

Line 79: remove “large” – this is a vague qualifier

Response new line 81: Accepted 

Line 84: the area of a circle with a 100m radii would be 0.0314km2

Response new line 88: Accepted 

Line 107: At what time of year did sampling occur in 2017? How many times were sites visited/resampled? What were the intervals of collection? How does this relate to known activity time of life stages in the region?

Response new line 113: Sampling for both years took place from June to October. At each site we sampled 5 or 10 plots and each site were visited once. During each collection day we sampled on average three sites sampling either from more natural to urban sites or in the direction from urban to natural sites, changing the direction randomly.

Ticks may have both unimodal and bimodal or even multimodal activity patters for the different tick stages (Mejlon, 2000). Therefore, we have sampled ticks from June to October for both 2017 and 2019. 

Line 127: At what time of year did sampling occur in 2019? How many times were sites visited/resampled? What were the intervals of collection? How does this relate to known activity time of life stages in the region?

Response new line 133: Please see previous comment 

Line 128: Were any of the sites sampled in 2017 resampled in 2019? It is unclear.

Response new line 134: We visited each site in 2017 and 2019 once, never resampled. 

Line 129: Okay. So, this statement makes it sound like each site was only sampled once over the entire duration of the study. That means a site consists of sampling from a single time point and 5 or 10 4m2 plots. This totals, at most, 40m2 of sampling per data point. While there is no authority over what the minimum sampling effort for ticks should be, this seems very low and may explain why so few adults were collected over the course of the study.

Response new line 136: Due to the longevity of I. ricinus ticks, a large amount of time is spent off the host, where the environment is posing a threat of survival of individual ticks (Mejlon, 2000). We believe that the tick life stage ratios in our study reflects the actual ratios in many areas and previous studies have found similar ratios with many larvae and nymphs and few adult ticks (Dobson et al., 2011; Ruiz-Fons et al., 2012). Due to the fact that we have sampled 5 – 10 plots per site the proportion of adult ticks is representative for the sample site, but very low in absolute numbers. Therefore, we agree with conclusion above from the reviewer. 

Line 135: It is unclear what local landcover the sampling occurred in. It is just referred to as greenspace here, but was it forest and if so what type? This is central to the second objective of the study.

Response new line 142: Since we have randomly distributed our sampling sites in Stockholm County, our sampling sites consists of a quite large variety of different greenspaces. To analyze the effects of the local habitat, we have used the smallest buffer zone (100m) in the GIS analyses. In addition, to clarify we have restated the objectives of this study.

Line 138: What is the snowball technique? Is that what is described in lines 138-144?

Response new line 144: Instead of using the term snowball technique we are instead describing the methods how we are randomly selecting the different plots at each site.

Line 154: How do you “cover” the activity patterns of ticks by only sampling each site a single time and having those single sampling occasions span from May to October?

Response new line 161: See answer to comment #5

Line 160: what is meant by the “height of the field layer”? what distance around the plot was used for stem density measurements? Was vegetation height/tree cover also measure in both 2017 and 2019, or only 2019?

Response: The height of the field layer is the average vegetation height in each plot. 

The tree cover density was measured using the Bitterlish technique which measures the number of tree stems adjusted for stem diameter at breast height and distance from the plot. In practice this means that we are counting small and large trees close to the plot but only large trees far away. The maximum distance for inclusion will differ depending in the size of the trees. 

Vegetation height and tree cover were measured both in 2017 and in 2019 for each sampling plot. 

Line 179: include list of the eight forest subcategories here

Response new line 187: We have added the eight forest subcategories used in this study to the methods section.

Line 182: This size of the forest fragments that sampling sites were in would be an important variable to include in this analysis.

Response new line 195: The effects of fragment size is included in the Contagion (CONTAG) variable.

Line 184: Justification for why 1000m buffers were used for urbanization and landscape composition models is needed. One could perform an analysis to determine the appropriate ‘scale of effect’.

Response: Landscape configuration is only important if individual habitat types is subdivided into several fragments or if the diversity of habitat types differs among sampling sites. These effects will most likely only be evident at the largest buffer zone size. 

Line 185: Was SHDI and contagion measured using the landcover classification with or without the forest subcategories?

Response new line 197: SHDI and contagion were measured with all forest categories collapsed into a single category FOREST. 

Line 193: do you mean “To analyze the effect of possible risk factors….”?

Response new line 203: Thank you pointing this out. We have revised the sentence. 

Line 210: It is unclear how spatial autocorrelation is accounted for within the models. Please explain and/or show results of tests for spatial autocorrelation.

Response new line 222: In the statistical models, site specificity (using site as random factor) accounts for the correlation that may exist between ticks and the environment that may create a bias in the result from repetitive sampling at each sampling site. We have revised the sentence changing from spatial autocorrelation to spatial correlation to be clearer. 

Line 216: What aspect/measure of each land cover type is being used? Proportion?

Response new line 229: Yes, we have used proportion of different land cover types in the different buffer zones. 

Line 216: How was this variation in sampling effort (5 vs 10 plots) accounted for in your response variable of abundance?

Response new line 239: By the inclusion of sampling site as a random factor in the mixed models

Line 219: Including the proportion of all landcover types in a single model is likely not appropriate as explanatory variables are very likely to be highly correlated given that the data is proportional. While the authors state that they tested for collinearity, these results are not reported, and the full models probably violated this rule. Please show VIF results and/or reduce variables included in full model.

Response new line 233: The models are first reduced based on the significance levels and AIC values. The reduced model fit was assessed by the residual distribution of each model using package DHARMa and lastly tested for collinearity. We have added in the manuscript which variables that had to be removed from the models since they were highly correlated. 

Line 239: Need justification for why counts of nymphs and adults were combined. Because few adults were collected, it seems reasonable to just run a model for larvae and a model for nymphs.

Response new line 255: The adults have been excluded from the statistical models. 

Line 277: Could the fewer ticks in taller vegetation be a detection issue?

Response new line 294: We believe that the fewer ticks in taller vegetation is a negative effect of vegetation height and not detection issue. With the mopping technique, the mop handle permit easy adjustments to different vegetation heights. See discussion in the manuscript

Table 1: Why so little consistency in significant landscape variables across scales for larval presence/absence and abundance?

Response: We have reanalyzed the larvae models and kept the factor time to be present in all models (even though not significant). The same procedures were done for the models with nymphs. As a result, we can now see more consistency in significant variables across scales. 

Line 449: It is confusing to have a discussion of tick abundance under a section title tick presence and absence.

Response new line 496: I think the title was tick presence and abundance.

Figure 3: Add an inset map for spatial context and explain the difference between the white and black in the background.

Response: Thank you for pointing this out. An inset map for the spatial context has been added as well as a legend for the black, white, and blue background. 

Thank you for all the time and comments from both the editor and reviewer. 

Kindest regards, 

Thérese Janzén 

References

Dobson AD, Taylor JL, Randolph SE. Tick (Ixodes ricinus) abundance and seasonality at recreational sites in the UK: hazards in relation to fine-scale habitat types revealed by complementary sampling methods. Ticks and tick-borne diseases. 2011 Jun 1;2(2):67-74.

Mejlon H. Host-seeking activity of Ixodes ricinus in relation to the epidemiology of Lyme borreliosis in Sweden (Doctoral dissertation, Acta Universitatis Upsaliensis). 

Ruiz-Fons F, Fernández-de-Mera IG, Acevedo P, Gortázar C, de la Fuente J. Factors driving the abundance of Ixodes ricinus ticks and the prevalence of zoonotic I. ricinus-borne pathogens in natural foci. Applied and environmental microbiology. 2012 Apr 15;78(8):2669-76.

---

## [Editor Report · Decision Letter 2]

14 Feb 2023

PONE-D-22-15726R2Factors responsible for *Ixodes ricinus* presence and abundance across a natural-urban gradientPLOS ONE

Dear Dr. Janzen,

Thank you for submitting your manuscript to PLOS ONE. After careful consideration, we feel that it has merit but does not fully meet PLOS ONE’s publication criteria as it currently stands. Therefore, we invite you to submit a revised version of the manuscript that addresses the points raised during the review process.

I feel you've done a generally good job of addressing the major concerns of previous reviewers.  The exclusion of adults from the analyses and re-focus on nymphs and larvae was necessary given so few adults were collected.  The design, sampling, and analysis are now better described and adequately justified.  However, points where clarity is lacking remain and the text in general needs tightened up.  Also, I think a focused paragraph needs to be added to the discussion that explicitly addresses how the lack of data for adults factors into our understanding of environment-tick-human encounters learned from this study.  For example, would you expect the same patterns or no and why or why not?  Are there any caveats to the conclusions given the lack of data on adults?  The Results section currently contains a fair amount of Materials and Methods.  Below I list a few specific examples, but this section needs to be gone through with close scrutiny such that the Results is only results.  In many of these cases, the text is redundant with what was already written in the Materials and Methods section.  Finally, clarity would be greatly improved if the Results could be restructured in some more meaningful way.  It currently reads like a laundry list of results, which makes it hard for readers to identify key, interesting patterns.  I'm not convinced every significant results needs to be mentioned.  Why not just make a general statement about factors influencing abundance/presence, cite out to the Tables, and then in the text only highlight those results that are particularly meaningful or revealing?  As far as I can tell, 10 plots/site were sampled in 2017 and 5 plots/site were sampled in 2019.  If I have that right, then for clarity simply just state that rather than using verbiage like '5 or 10 were sampled'.What factors are local site/scale factors?  In the M&Ms (236-237) you list vegetation height, tree stem density, and temperature.  But then in Table 4, you indicate coniferous forests. Could Tables 1&2 be divided into local scale factors vs all others? Do these Tables only include factors that are significant? I'm assuming so, so please explicitly state that in the caption. Numbering and discussion order of Tables & Figures should correspond. Currently, you cite out to Table 4 before Table 3. Either change the order of their discussion or the numbers of the tables.In the text *Time *is identified as a factor but the Tables don't list *Time* as a factor, they list *Month*.  Choose either Time or Month.Regularity of sampling is still unclear.  Did you sample weekly? bi-weekly? daily? 3 days in a row? some other way?  Please clarify.Line 70: N. Europe is a misnomer given you only looked at one spot in N. Europe, i.e. Stockholm CountyLine 84: Delete the period after 'sites'Line 105: Change 'picture' to 'figure'There are many spots where verb agreement is incorrect, e.g. Line 105: 'show' should be 'shows', line 135 'were' should be 'was', line 137 'allow' should be 'allows', line 222 'account' should be 'accounts', line 291 'was' should be 'were', line 316 'represents' should be 'represent', etc. Given how many I uncovered just quickly reading through certain sections, I'm guessing there are others. Please carefully scrutinize all text to ensure verb usage is correct.Line 130: Insert 'them' after 'excluded'Line 222: delete either 'these' or 'the'Lines 232-238: This is a really long sentence.  Rather than having 'fixed factors' at the end, how about flipping the text around to something like, Fixed factors included...  Then list the factors.Line 239: Add 'a' before 'random'Line 256: Replace 'but not for adult ticks' with 'and excluded adult ticks from the analyses'.Line 273: Replace existing with, "Adult ticks were excluded from all statistical analyses". (I was back and forth on whether this should only be here or only in M&Ms or in both, and decided it seemed best in both locations)Lines 282-284: Delete and add as footnote to the Tables.Line 285: 'provide' should be 'provided'Lines 290-292: If these were the only local factors (as indicated in lines 236-237), then how about just simply state something like, All local site factors, i.e. veg height, tree stem density, and temp, all had a significant effect on...Examples of M&Ms included in the Results: Lines 273-274, 280-281, 292-295, 314-316, etc. Please carefully scrutinize the Results section and ensure that M&Ms are in the M&Ms section where they belong.Line 275-277: Since 273-274 will be removed since it's not results, simply reword to something like, Highly correlated variables removed from the analyses of landscape configuration...include...  etc.Line 318: 'lager' should be 'larger'.Please submit your revised manuscript by Mar 31 2023 11:59PM. If you will need more time than this to complete your revisions, please reply to this message or contact the journal office at plosone@plos.org. Please include the following items when submitting your revised manuscript:

We look forward to receiving your revised manuscript.

Kind regards,

Janice L. Bossart

Academic Editor

PLOS ONE
---

## [Author Response · Author response to Decision Letter 2]

24 Mar 2023

Dear Dr. Bossart,

Thank you for letting us resubmit our manuscript to PLOS ONE. Your comments were highly insightful and have improved our manuscript. Attached are our responses to your comments. 

We have added a paragraph to the discussion that addresses how the lack of data for adult ticks’ factors into our understanding of Ixodes ricinus presence and abundance across the urbanization gradient. In our study, adult ticks were absent in highly urbanized areas. Urbanized greenspaces that are surrounded by build environments generally have a reduced diversity of host animals, especially larger mammals on which adults’ feed. The few adult ticks collected in this study were sampled in suburban forests and at these sites other tick life-stages were also found. These suburban forests have favorable habitats for ticks and hosts and here all tick life stages can find a host for their blood-meals. 

We have also reworked the results section which is now presented with interesting patterns explaining I. ricinus presence and abundance. Lastly, we have worked through the manuscript, which is now tightened up and clear and to the point. All changes to the manuscript have been marked with yellow color. 

Below is our point-by-point response to each of your comments.

On behalf of all the authors

Sincerely,

Thérese Janzén

Corresponding author

As far as I can tell, 10 plots/site were sampled in 2017 and 5 plots/site were sampled in 2019. If I have that right, then for clarity simply just state that rather than using verbiage like 'Line 245-249: were sampled'.

Response: Accepted

What factors are local site/scale factors? In the M&Ms (236-237) you list vegetation height, tree stem density, and temperature. But then in Table 4, you indicate coniferous forests. Could Tables 1&2 be divided into local scale factors vs all others? Do these Tables only include factors that are significant? I'm assuming so, so please explicitly state that in the caption. 

Response: We have divided up table 1 and 2 in local site factors (100m) and surrounding landscape (200-1000m). Table 4 we have divided into local plot, local site and surrounding landscape. 

Numbering and discussion order of Tables & Figures should correspond. Currently, you cite out to Table 4 before Table 3. Either change the order of their discussion or the numbers of the tables.

Response: Accepted

In the text Time is identified as a factor but the Tables don't list Time as a factor, they list Month. Choose either Time or Month.

Response: Accepted

Regularity of sampling is still unclear. Did you sample weekly? bi-weekly? daily? 3 days in a row? some other way? Please clarify.

Response: Our sampling has been irregular and therefore we have added the time factor Month into our models. We have added information regarding the regularity of our sampling to our manuscript. 

Line 135: Field sampling for both years occurred irregularly with most sampling times in June and August.

Line 245-249: The second order polynomial of Month indicating inventory time (sampling Day caused convergence issues) was added to correct for the irregular sampling times. In addition, sampling site was used as random variable to adjust for the difference in sampling effort between sites, and to handle the correlative structure among plots within the same sampling site in the conditional part of the model.

Line 70: N. Europe is a misnomer given you only looked at one spot in N. Europe, i.e. Stockholm County

Response: Accepted

Line 84: Delete the period after 'sites'

Response: Accepted

Line 105: Change 'picture' to 'figure'

Response: Accepted

There are many spots where verb agreement is incorrect, e.g. Line 105: 'show' should be 'shows', line 135 'were' should be 'was', line 137 'allow' should be 'allows', line 222 'account' should be 'accounts', line 291 'was' should be 'were', line 316 'represents' should be 'represent', etc. Given how many I uncovered just quickly reading through certain sections, I'm guessing there are others. Please carefully scrutinize all text to ensure verb usage is correct.

Response: We have carefully scrutinized all text. 

Line 130: Insert 'them' after 'excluded'

Response: Accepted

Line 222: delete either 'these' or 'the'

Response: Accepted

Lines 232-238: This is a really long sentence. Rather than having 'fixed factors' at the end, how about flipping the text around to something like, Fixed factors included... Then list the factors.

Response: Accepted

Line 239: Add 'a' before 'random'

Response: Accepted

Line 256: Replace 'but not for adult ticks' with 'and excluded adult ticks from the analyses'.

Response: Accepted

Line 273: Replace existing with, "Adult ticks were excluded from all statistical analyses". (I was back and forth on whether this should only be here or only in M&Ms or in both, and decided it seemed best in both locations)

Response: Accepted

Lines 282-284: Delete and add as footnote to the Tables.

Response: Accepted

Line 285: 'provide' should be 'provided'

Response: Accepted

Lines 290-292: If these were the only local factors (as indicated in lines 236-237), then how about just simply state something like, All local site factors, i.e. veg height, tree stem density, and temp, all had a significant effect on...

Response: Accepted

Examples of M&Ms included in the Results: Lines 273-274, 280-281, 292-295, 314-316, etc. Please carefully scrutinize the Results section and ensure that M&Ms are in the M&Ms section where they belong.

Response: Accepted

Line 275-277: Since 273-274 will be removed since it's not results, simply reword to something like, Highly correlated variables removed from the analyses of landscape configuration...include... etc.

Response: Accepted

Line 318: 'lager' should be 'larger'.

Response: Accepted

---

## [Editor Report · Decision Letter 3]

3 Apr 2023

PONE-D-22-15726R3Factors responsible for *Ixodes ricinus* presence and abundance across a natural-urban gradientPLOS ONE

Dear Dr. Janzen,

Thank you for submitting your manuscript to PLOS ONE. After careful consideration, we feel that it has merit but does not fully meet PLOS ONE’s publication criteria as it currently stands. Therefore, we invite you to submit a revised version of the manuscript that addresses the points raised during the review process.

 The changes you have made have definitely improved manuscript organization and clarity.  Unfortunately, as I read through I'm continuing to find additional errors.  PLOS ONE does not rely on copy editors.  The onerous for ensuring the manuscript is as error free as possible lies with the authors.  Below I've listed errors I noted just reading through the yellow highlighted text.  Given the number found, I'm not at all confident that aren't many additional errors throughout the manuscript.  Since it is easy to miss errors when one is very familiar with a manuscript, please have someone who is unfamiliar with your manuscript read through it slowly and closely to ensure your submission is as error free as possible. Line 84-85.  Delete the 2nd 'in 2019'Line 199. Add a comma after 'attributes'Line 278. Remove 'that'Line 284. 'Curve linear' should be 'curvilinear'Line 307. Capitalize 'spruce' for consistencyLine 403. 'dryer' should be 'drier'Line 406. 'steam' should be 'stem'Line 465. Add a comma after 'stage'Line 467. 'build' should be 'built'Table 2 Heading. 'Ricinus' should be lower caseTable 4 Heading. Should be 'Landscape features include...' Please italicize all scientific names.  The vast majority of these are *Ixodes ricinus*, but there are a few others. Please submit your revised manuscript by May 18 2023 11:59PM. If you will need more time than this to complete your revisions, please reply to this message or contact the journal office at plosone@plos.org. Please include the following items when submitting your revised manuscript:A rebuttal letter that responds to each point raised by the academic editor and reviewer(s). You should upload this letter as a separate file labeled 'Response to Reviewers'.A marked-up copy of your manuscript that highlights changes made to the original version. You should upload this as a separate file labeled 'Revised Manuscript with Track Changes'.An unmarked version of your revised paper without tracked changes. You should upload this as a separate file labeled 'Manuscript'.If applicable, we recommend that you deposit your laboratory protocols in protocols.io to enhance the reproducibility of your results. Protocols.io assigns your protocol its own identifier (DOI) so that it can be cited independently in the future. For instructions see: https://journals.plos.org/plosone/s/submission-guidelines#loc-laboratory-protocols. Additionally, PLOS ONE offers an option for publishing peer-reviewed Lab Protocol articles, which describe protocols hosted on protocols.io. Read more information on sharing protocols at https://plos.org/protocols?utm_medium=editorial-email&utm_source=authorletters&utm_campaign=protocols.

We look forward to receiving your revised manuscript.

Kind regards,

Janice L. Bossart

Academic Editor

PLOS ONE
---

## [Author Response · Author response to Decision Letter 3]

26 Apr 2023

Dear Dr. Bossart,

Thank you for letting us resubmit our manuscript to PLOS ONE. We took your advice and asked someone who was unfamiliar with our manuscript and who works as an academic writer read it through carefully. All changes to the manuscript have been marked with yellow color. 

Below is our point-by-point response to each of your comments.

On behalf of all the authors

Sincerely,

Thérese Janzén

Corresponding author

Line 84-85. Delete the 2nd 'in 2019'

Response: Accepted

Line 199. Add a comma after 'attributes'

Response: Accepted

Line 278. Remove 'that'

Response: Accepted

Line 284. 'Curve linear' should be 'curvilinear'

Response: Accepted

Line 307. Capitalize 'spruce' for consistency

Response: Accepted

Line 403. 'dryer' should be 'drier'

Response: Accepted

Line 406. 'steam' should be 'stem'

Response: Accepted

Line 465. Add a comma after 'stage'

Response: Accepted

Line 467. 'build' should be 'built'

Response: Accepted

Table 2 Heading. 'Ricinus' should be lower case

Response: Accepted

Table 4 Heading. Should be 'Landscape features include...'

Response: Accepted

Please italicize all scientific names. The vast majority of these are Ixodes ricinus, but there are a few others.

Response: Accepted

---

## [Editor Report · Decision Letter 4]

3 May 2023

Factors responsible for *Ixodes ricinus* presence and abundance across a natural-urban gradient

PONE-D-22-15726R4

Dear Dr. Janzen,

We’re pleased to inform you that your manuscript has been judged scientifically suitable for publication and will be formally accepted for publication once it meets all outstanding technical requirements. Congratulations!

Typically, changes between revisions should only include those requested by external reviewers and the editors.  Your Revision 4 text differs substantially from that of Revision 3.  My suggestion to have someone unfamiliar with the manuscript read through was only meant with respect to finding spelling/grammatical errors, not to result in substantial modifications to the text and its structure/organization.  Typically significant modifications require additional review.  That said, in reading through the many text modifications this most recent revision incorporated, I recognize they only deal with wording and flow have made the manuscript easier to read and follow.

I note that scientific names are still not capitalized within the reference section.  Please italicize all scientific names prior to submitting the final version for publication.

Kind regards,

Dr. Janice L. Bossart

Academic Editor

PLOS ONE
---

## [Editor Report · Acceptance letter]

8 May 2023

PONE-D-22-15726R4 

Factors responsible for *Ixodes ricinus* presence and abundance across a natural-urban gradient 

Dear Dr. Janzén:

I'm pleased to inform you that your manuscript has been deemed suitable for publication in PLOS ONE. Congratulations! Your manuscript is now with our production department. 

Kind regards, 

on behalf of

Dr. Janice L. Bossart 

Academic Editor

PLOS ONE